

# 1 Monoterpene chemical speciation in the Amazon tropical rainforest: variation with season, height, and time of day at the Amazon Tall Tower Observatory (ATTO)

**Ana María Yañez-Serrano[1,*], Anke Christine Nölscher[1], Efstratios Bourtsoukidis[1], Eliane Gomes Alves[2], Laurens Ganzeveld[3], Boris Bonn[4], Stefan Wolff[1], Marta Sa[2], Marcia Yamasoe[5], Jonathan Williams[1], Meinrat O. Andreae[1,6] and Jürgen. Kesselmeier[1]**

[1]{ Air Chemistry, Biogeochemistry and Multiphase Departments, Max Planck Institute for Chemistry, D-55020 Mainz, Germany}

[2]{Instituto Nacional de Pesquisas da Amazônia (INPA), Av. André Araújo 2936, Manaus-AM, Brazil}

[3] {Meteorology and Air Quality (MAQ), Department of Environmental Sciences, Wageningen University and Research Centre, Wageningen, the Netherlands}

[4] {Chair of Tree Physiology, Albert Ludwig University, Freiburg, Germany}

[5] {Instituto de Astronomia, Geofísica e Ciências Atmosféricas, Universidade de São Paulo, Rua do Matão 122, São Paulo-SP, Brazil}

[6] {Scripps Institution of Oceanography, University of California San Diego, La Jolla, CA, USA}

* Now at Chair of Ecosystem Physiology, Albert Ludwig University, Freiburg, Germany

Correspondence e-mail: a.yanezserrano@mpic.de

Speciated monoterpene measurements in the Amazon rainforest air are scarce, but important in order to understand their contribution to the overall reactivity of volatile organic compound (VOCs) emissions towards the main atmospheric oxidants, such as hydroxyl radical (OH), ozone ($O_3$) and nitrate radical ($NO_3$). In this study, we present the chemical speciation of gas phase monoterpenes measured in the tropical rainforest at the Amazon Tall Tower Observatory (ATTO, Amazonas, Brazil). Samples of VOCs were collected by two automatic sampling systems positioned on a tower at 12 and 24 m height and analysed using Gas Chromatography Flame Ionization Detection (GC-FID). The samples were collected in October 2015, representing the dry season, and compared with previous wet and dry season studies at the site. In addition, vertical profile measurements (at 12 and 24 m) of total monoterpene mixing ratios were made using Proton-Transfer Reaction Mass Spectrometry (PTR-MS). The results showed a distinctly different chemical speciation between day and night. For instance, α-pinene was more abundant during the day, whereas limonene was more abundant at night. Reactivity calculations showed that the most abundant compounds may not be the most atmospheric chemically relevant compounds. Furthermore, inter- and intra-annual results demonstrate similar chemodiversity during the dry seasons analysed. Simulations with a canopy exchange modelling system compare relatively well with the observed temporal variability in speciated monoterpene mixing ratios, but also indicate the necessity of more experiments to enhance our understanding of in-canopy sinks of these monoterpenes.

## 1. Introduction



Isoprenoids such as isoprene ($C_5H_8$), monoterpenes ($C_{10}H_{16}$) and sesquiterpenes ($C_{15}H_{24}$) are
considered to be key contributors to the production of biogenic secondary organic aerosol (SOA), which
affect cloud condensation nuclei production (Engelhart et al., 2008; Jokinen et al., 2015; Pöschl et al.,
2010). While isoprene has been shown to be a globally significant source of SOA (Claeys et al., 2004),
it's presence has been also shown to inhibit SOA formation under certain conditions (Kiendler-Scharr et
al., 2009). By virtue of their lower volatility and higher ozone reactivity, monoterpenes and sesquiter-
penes are strong sources of secondary organic aerosol (SOA) through the generation of low-volatility
oxidation products formed via ozonolysis and hydroxyl radical oxidation (Bonn and Moortgat, 2003;
Zhao et al., 2015).

The main source of monoterpenes to the global atmosphere is emission from vegetation, with
smaller contributions from soil (Kesselmeier and Staudt, 1999; Kuhn et al., 2002; Ormeno et al., 2007).
Synthesis of the monoterpene species occurs via the non-mevalonate pathway within the plant chloro-
plast (Kesselmeier and Staudt, 1999; Lichtenthaler, 1999; Schwender et al., 1996), which explains the
light dependency also known to determine isoprene synthesis and emission. These commonly emitted
compounds have been identified as important signalling compounds through plant-to-plant, plant-insect
or plant-microbe interactions (Gershenzon, 2007; Gershenzon and Dudareva, 2007; Kishimoto et al.,
2006; Maag et al., 2015) and they are thought to protect photosynthetic membranes against abiotic
stresses (Jardine et al., 2017; Penuelas and Llusia, 2002; Vickers et al., 2009).

Despite having a common sum formula, variations in the molecular structure of the various
monoterpenes result in large variations (over two orders of magnitude) of their reaction rate coefficients
with the hydroxyl radical (OH), ozone ($O_3$) and nitrate radical ($NO_3$). Current quantification methods
for monoterpenes include the fast (ca. 30 seconds) but unspecific total monoterpene measurement by the
proton transfer reaction  mass spectrometer (PTR-MS) (Lindinger and Jordan, 1998) and the slower (ca.
1 hour) but chemically speciated gas chromatographic methods. In order to gauge the role of these spe-
cies in atmospheric chemistry, chemical speciation needs to be provided by the gas chromatographic
techniques while the on-line mass spectrometer can assess how the total mass changes over time. It



should be noted that the structure and reactivity of the monoterpenes can also have implications for the
efficiency of SOA formation (Hallquist et al., 2009; Kiendler-Scharr et al., 2009; Mentel et al., 2009;
O'Dowd et al., 2002). In most cases, SOA products are poorly characterized due to a scarcity of meas-
urements (Martin et al., 2010).

Considering the overall size of the Amazon rainforest (5.4 million $km^2$ in 2001; Malhi et al.,
2008) and the significant contribution of BVOC emissions from this vast forest to the global VOC
budget (globally 1000 Tg of carbon $yr^{-1}$; Guenther et al., 2012), measurements of total monoterpene
emissions and mixing ratios from this ecosystem are scarce (Greenberg and Zimmerman, 1984; Helmig
et al., 1998; Jardine et al., 2015, 2011, 2017; Karl et al., 2007; Rinne et al., 2002; Yáñez-Serrano et al.,
2015). Speciated measurements are even more scarce (Jardine et al., 2015, 2017; Kesselmeier et al.,
2002; Kuhn et al., 2004). Yet, this information is essential to our understanding of the functioning of the
Amazon rainforest in atmospheric chemistry-climate interactions. Knowledge of these processes also
serves to improve predictions of future changes in atmospheric composition and to assess the impact of
changes in regional emissions and land use on global climate caused by Amazon deforestation.

In this study, we evaluate measurements of speciated rainforest monoterpene mixing ratios as a
function of height in the canopy, season and diel cycle. This evaluation includes a comparison with an
offline version of the Multi-Layer Canopy Chemistry Exchange model (MLC-CHEM), driven by ob-
served micro-meteorology and ozone surface layer mixing ratios, to support analysis of the measured
temporal variability in speciated rainforest monoterpene mixing ratios as a function of height in the can-
opy. The MLC-CHEM was chosen for this purpose since it has been already extensively applied for
site- to global-scale studies on atmosphere-biosphere exchange for tropical rainforest ecosystems
(Ganzeveld et al., 2008, 2002a; Ganzeveld and Lelieveld, 2004; Kuhn et al., 2010).

**2.  Methodology**
2.1. Site





The site chosen for this study was the Amazon Tall Tower Observatory, ATTO (Andreae et al., 2015).
This site is located in Central Amazonia (S 02° 08.647' W 58° 59.992'), 150 km NE of the closest pop-
ulated city, Manaus, Brazil. Due to the prevailing north-easterly wind direction, the influence of the Ma-
naus plume is negligible and the measurements at this site can be considered to reflect pristine tropical
forest conditions affected by air masses that passed over 1000 km of undisturbed rainforest. The site is
equipped with a 325 m tall tower as well as two smaller towers. This study was carried out on the IN-
STANT tower, an 80-m walk-up tower located 600 m from the tall tower in easterly direction. For a
comprehensive site description see Andreae et al. (2015). Sampling was performed on this tower at two
different heights (12 m and 24 m).

2.2. Air sampling

Collection of ambient air samples on adsorbent tubes, for subsequent analysis by Gas Chroma-

tography – Flame Ionization Detector (GC-FID), was made with two automated cartridge samplers de-
scribed earlier (Kesselmeier et al., 2002; Kuhn et al., 2002, 2005). The samples were collected from 17
to 20 October 2015. Furthermore, additional sampling was performed at 24 m with a GSA SG-10-2 per-
sonal sampler pump during the years 2012-2014. Measurements took place on 19 and 28 November
2012, 22, 25 and 26 September 2013, 17 and 21 August 2014, with the measurements presented in this
study in more detail having been taken from 17 to 20 October 2015 (dry season); 1, 3 and 4 March 2013
(wet season) and 11 to 14 June 2013 (wet-to-dry transition). Briefly, the samplers consist of two main
units, a cartridge magazine that holds the adsorbent-filled tubes and the control unit timing the process
and recording the data. This latter unit also houses the pumps (Type N86KT, KNF Neuberger, Freiburg,
Germany), pressure gauges, mass flow controllers and power supply. The cartridge magazine is
equipped with solenoid valves controlling the inlet and outlet of up to 20 individual sampling adsorbent
tubes. The system is a constant-flow device, with one cartridge position per loop used as a bypass for
purging the system. Due to the compact weatherproof housings and the low power consumption, we
were able to position one sampler at 24 m and the other one at 12 m, attached to the INSTANT tower
booms with commercially available 50 mm aluminium clamps. The height of the canopy is approxi-
mately 35 m (Andreae et al., 2015).




The adsorbent tubes used for VOC sampling were filled with 130 mg of Carbograph 1 (90 m$^2$ g$^{-1}$)
followed by 130 mg of Carbograph 5 (560 m$^2$ g$^{-1}$) sorbents. The size of the Carbograph particles was
in the range of 20–40 mesh. Carbographs 1 and 5 were provided by L.A.R.A s.r.l. (Rome, Italy)
(Kesselmeier et al., 2002). Samples were taken for 30 min at a flow of 200 cm$^3$ min$^{-1}$ (STP), leading to
a collection of 6 L of air.

2.3.  Instruments for chemical analysis

2.3.1.   Gas Chromatography –Flame Ionization Detector (GC-FID)

After collection, the adsorbent tubes were analysed at the Max Planck Institute for Chemistry
(MPIC) using a Gas Chromatograph equipped with a Flame Ionization Detector (GC-FID, Model Auto-
System XL, Perkin Elmer GmbH, Germany) for identification and quantification of the monoterpene
species. Helium was used as carrier gas. Separation occurred on a 100 meter HP-1 column with 0.22
mm inner diameter and coated with the non-polar dimethylpolysiloxane as stationary phase. The com-
pound mixture collected in the adsorbent tubes was discharged into the gas stream with the help of a
two-step desorption system (Model ATD400, Perkin Elmer, Germany). The samples were cryofocused
in a cold trap at -30 °C filled with Carbograph 5, providing better defined peaks in the chromatograms.
Afterwards the cold trap was rapidly heated to 280°C and the pre-concentrated sample injected onto the
column. The following temperature programme was used: (-10 to 40 °C at 20 °C min$^{-1}$, 40 to 145 °C at
1.5 °C min$^{-1}$, and 145 to 220 °C at 30 °C min$^{-1}$). The separated compounds were quantified with a
Flame Ionization Detector (FID). Identification was achieved through spiked injection of pure com-
pounds. For a more detailed description see Kesselmeier et al., (2002).

Calibration for VOCs containing no heteroatoms was achieved by using a standard gas mixture
of isoprene and several n-alkanes (n-pentane, n-hexane, n-heptane, n-octane, n-nonane, and n-decane)
(Apel-Riemer Environmental Inc., USA). In this case, it is assumed that the "effective carbon number"
(Sternberg et al., 1962) is equal to the real carbon number of the molecules (Komenda, 2001), yielding a



signal response that is proportional to the real carbon number. The monoterpenes identified and quanti-
fied were α-pinene, camphene, sabinene, β-pinene, myrcene, α-phellandrene, 3-carene, α-terpinene, ρ-
cymene, limonene and γ-terpinene. The detection limit for the GC-FID was 2 ppt (Bracho-Nunez et al.,

2011).


### 2.3.2. Proton Transfer Reaction - Mass Spectrometer (PTR-MS)
Online total monoterpene mixing ratios were determined by a quadrupole Proton Transfer Reac-
tion - Mass Spectrometer, PTR-MS (Ionicon Analytic, Austria). The PTR-MS was operated under
standard conditions (2.2 mbar drift pressure, 600 V drift voltage, 142 Td). Periodic background meas-
urements and humidity dependent calibrations were performed. A gravimetrically prepared multicom-
ponent standard for calibration was obtained from Apel & Riemer Environmental, USA. The measure-
ments were carried out at eight different heights (0.05, 0.5, 4, 12, 24, 53 and 79 m) with the PTR-MS
switching sequentially between each height at 2 min intervals and only data from 12 and 24 m is shown.
The inlet lines were made of PTFE (9.5 mm OD), insulated and heated to 50 ºC, and had PTFE particle
inlet filters at the intake end. More information about the gradient system and PTR-MS operation at
ATTO can be found elsewhere (Nölscher et al., 2016; Yáñez-Serrano et al., 2015). The limit of detec-
tion (LOD) of the PTR-MS for total monoterpenes was 0.1 ppb, determined as 2σ of the background
noise.

### 2.4. Multi-Layer Canopy Chemistry Exchange model (MLC-CHEM)
To analyse the magnitude and temporal variability in the observed monoterpene concentrations
inside and above the forest canopy, we applied the Multi-Layer Canopy Chemistry Exchange Model
(MLC-CHEM), driven by the observed micro-meteorology and ozone surface layer mixing ratios. The
MLC-CHEM model was originally developed and implemented as a single-column model (SCM) as
well as in a global chemistry and climate-modelling system to assess the role of canopy processes in lo-
cal- to global-scale atmosphere-biosphere exchange of nitrogen oxides (Ganzeveld et al., 2008, 2002b;
Kuhn et al., 2010). The MLC-CHEM generalized representation of chemistry, dry deposition, emissions



and turbulent mixing allows to study the role of canopy interactions in determining atmosphere-bio-
sphere exchange fluxes and in-canopy and surface layer concentrations of biogenic volatile organic
compounds (BVOCs). BVOC emissions are calculated according to MEGAN (Guenther et al., 2006),
considering the vertical distribution of biomass and direct as well as diffuse radiation to calculate leaf-
scale BVOC emissions. The current implementation of canopy chemistry in MLC-CHEM considers, in
addition to standard photo-chemistry ($O_3$, $NO_x$, $CH_4$, $CO$), the role of non-methane hydrocarbons in-
cluding isoprene, and a selection of hydrocarbon oxidation products such as formaldehyde, higher alde-
hydes and acetone. Oxidation of the monoterpenes by $OH$, $O_3$ and $NO_3$ is taken into account to simulate
their chemical destruction, but the role of the oxidation products in photo-chemistry is not considered in
the current implementation of the chemistry scheme in MLC-CHEM. For this study, we have extended
MLC-CHEM to consider, besides the already included compounds α-pinene and β-pinene, the observed
monoterpene species α-terpinene, limonene and myrcene. The monoterpene relative emission ratios
used for this simulation (relative to the selected initial leaf-scale MT emission factor (EA) of 0.4 µg C
$g^{-1}$ $hr^{-1}$) were: α-pinene = 0.45 x EA, β-pinene = 0.10 x EA, α-terpinene = 0.27 x EA, limonene = 2.25 x
EA and myrcene = 0.45 x EA. Note that for limonene, only the selected high basal emission flux would
result in simulated mixing ratios comparable to the observed ones. Regarding the physical sinks; dry
deposition of gases including the BVOC compounds depends on their uptake resistances calculated ac-
cording to Wesely's (1989) parameterization, which estimates these uptake resistances based on the
compounds' solubility and reactivity.

The simulations with MLC-CHEM were constrained with the observed surface layer net radia-

tion, wind speed, relative humidity and $O_3$ concentrations as well as the temperature measured above
and inside the canopy from 17 to 20 October 2015. Note that unfortunately no high-quality micromete-
orological observations were available for 18 October 2015, so we modelled a diurnal cycle based on
the previous and subsequent meteorological parameters, which drive MLC-CHEM quite close to the ob-
served diel cycles for the other three days of the measurement period.

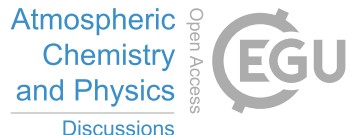

## 3. Results

### 3.1. Time series and diel cycles

Continuous online PTR-MS measurements were compared with off-line GC-FID samples over the course of 3 days in October 2015 (Figure 1). The close agreement between the two measurement techniques provides confidence that almost all monoterpenes present at ambient air in the site were being measured. Note that in this comparison, $\rho$-cymene was removed from the calculations as the PTR-MS does not detect it on m/z 137.





Figure 1: Graph showing the speciated monoterpene mixing ratios measured hourly from 17 to 20 October 2015 for b) 24 m and c) 12 m. The colours on the stacked bar plot indicate the different monoterpene species as they are denoted in the legend. The black line represents the PTR-MS total monoterpene mixing ratio, with a gap of data on the 19 October 2015. Temperature at 80 m is shown in the red thick line and photosynthetically active radiation at 39 m is shown in the shaded areas in a).

The total monoterpene mixing ratios were higher during the day when temperature and solar radiation were at their maxima. Most of the observed distinct diurnal cycle in total monoterpene mixing ratios can be attributed to α-pinene, which was the dominant species during the daytime with mixing ratios large as 0.33±0.04 and 0.38±0.21 ppb at 12 and 24 m respectively, and 0.15±0.05 and 0.11±0.06 ppb for the night at 12 and 24 m. The second most abundant monoterpene species was limonene, with





observed average daytime mixing ratios of 0.18±0.09 and 0.19±0.12 ppb at 12 and 24 m, respectively,
and 0.18±0.01 and 0.14±0.07 ppb for the night time at 12 and 24 m.

Table 1: Mixing ratios in ppb at 24 and 12 m of the measured monoterpene species for 17 to 20 October 2015 as determined
by the GC-FID analysis. The daytime period was considered from 0900h to 1700h and the night time period from 2000h to
0500h (Local time). BLD stands for below detection limit.

| Compound | Day 12 m | Night 12 m | Day 24 m | Night 24 m |
|---|---|---|---|---|
| α-Pinene | 0.33 ± 0.04 | 0.15±0.05 | 0.38±0.21 | 0.11±0.06 |
| Limonene | 0.18±0.09 | 0.18±0.10 | 0.19±0.12 | 0.14±0.07 |
| Myrcene | 0.16±0.14 | 0.12±0.09 | 0.09±0.04 | 0.07±0.06 |
| P-Cymene | 0.07±0.03 | 0.04±0.01 | 0.08±0.04 | 0.04±0.02 |
| β-Pinene | 0.08±0.03 | 0.06±0.03 | 0.05±0.03 | 0.04±0.02 |
| Camphene | 0.03±0.03 | 0.02±0.01 | 0.03±0.02 | 0.01±0.01 |
| α-Terpinene | 0.03±0.02 | 0.03±0.02 | 0.01±0.02 | 0.02±0.02 |
| γ-Terpinene | 0.02±0.01 | 0.01±0.01 | 0.01±0.01 | 0.01±0.01 |
| 3-Carene | 0.001±0.003 | 0.003±0.008 | 0.003±0.011 | 0 or BLD |
| α-Phellandrene | 0 or BLD | 0 or BLD | 0 or BLD | 0 or BLD |
| Sabinene | 0 or BLD | 0 or BLD | 0 or BLD | 0 or BLD |
| MT Sum – GC-FID | 0.91±0.10 | 0.62±0.19 | 0.82±0.34 | 0.45±0.13 |
| MT Sum – PTR-MS | 0.96±0.27 | 0.54±0.17 | 0.77±0.22 | 0.56±0.16 |


The difference between the 12 and 24 m height total monoterpene mixing ratios is minor given
the variance of the measurements, but there is a tendency for the difference to be more pronounced dur-
ing night time (Table 1). This could be due to higher biogenic emissions at 12 m compared to 24 m dur-
ing the night, differences in reactivity within the canopy leading to different oxidation regimes, or more
stagnant conditions leading to higher accumulation.

The continuous online measurements by the quadrupole PTR-MS indicate a clear diurnal cycle
in the measured mixing ratios of the sum of monoterpenes, which has been reported previously from
this site (Yáñez-Serrano et al., 2015). In order to assess the effect of each individual monoterpene spe-
cies, we further investigated their diurnal cycles as obtained by the off-line GC-FID samples. The meas-
ured diel cycles for the most relevant monoterpene species at the ATTO site were very similar at both





heights. We also compared the measured diel cycles of isoprene with the observed diel cycle for the dif-
ferent monoterpene species. The compounds that showed a diurnal cycle similar to isoprene were α-pi-
nene and ρ-cymene (Figure 2). This could be due to the emission of α-pinene and ρ-cymene being de-
pendent on light and temperature, analogous to isoprene. However, during the night both monoterpenes
were also present, albeit at lower mixing ratios, and the nocturnal mixing ratios of the monoterpenes did
not decrease as much as isoprene. This has also been noted in previous studies (Yáñez-Serrano et al.,

2015).


Despite the higher mixing ratios of limonene compared to other monoterpene species (other than
α-pinene), it was not possible to distinguish any clear diel pattern in the average data for this species
(see Figure 2). β-Pinene and α-terpinene likewise show no obvious diel pattern in the rainforest air, de-
spite the detection limit of the GC-FID being 2 ppt.

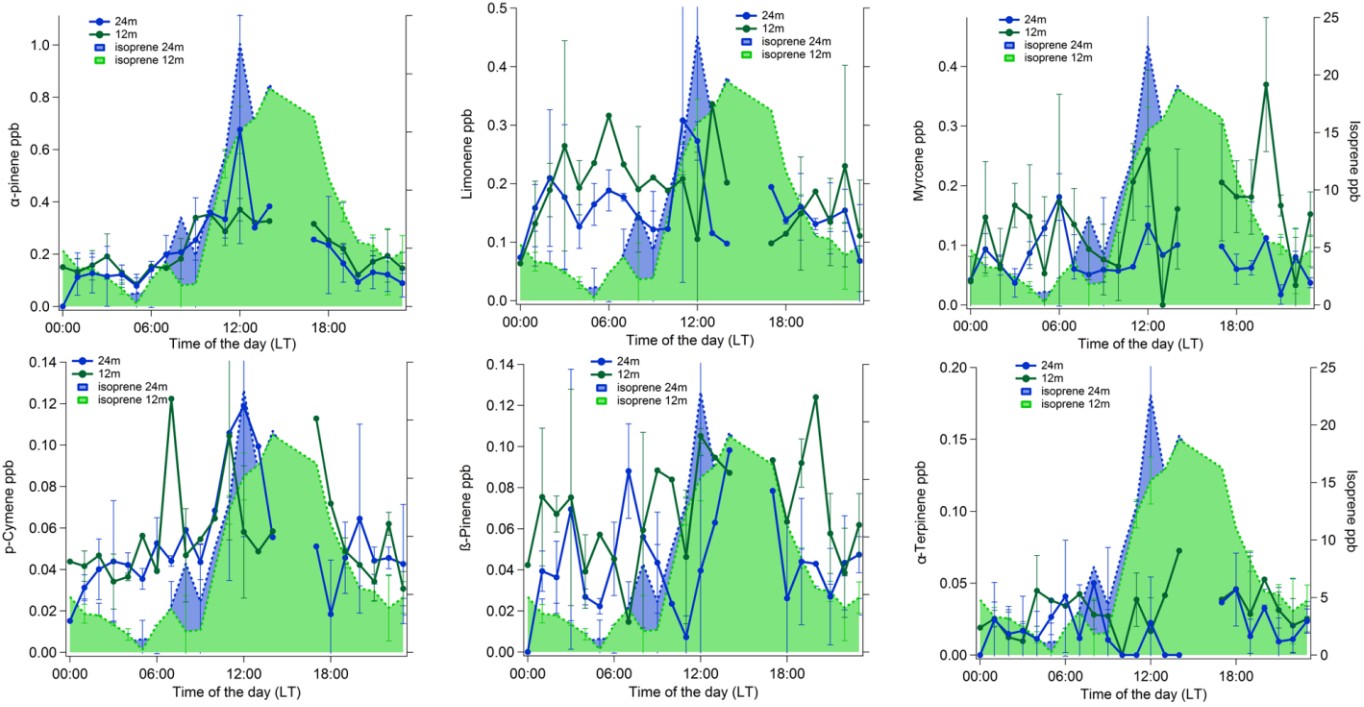

Figure 2: Graphs showing the average diel cycles for 24 m (in blue) and for 12 m (in green) for α-pinene, limonene, myr-
cene, ρ-cymene, β-pinene and α-terpinene, individually. All graphs have the isoprene diel cycle for both heights in the back.



### 3.2. Chemodiversity

The chemical speciation (or chemodiversity) of monoterpenes relates to the relative abundances of the different monoterpene species in the sampled air. α-Pinene, limonene, myrcene, ρ-cymene and β-pinene represented more than 85% of the total MT mixing ratio (Figure 3). During the day, α-pinene had an average abundance of 50% and 36% of the total monoterpene mixing ratios at 24 and 12 m, respectively, and was the dominant monoterpene in this study overall. However, during the night, its relative abundance dropped to 32% and 29% at 24 and 12 m, respectively. In contrast, limonene made up 19% and 20% of the MT at 24 and 12 m, respectively, by day, and increased during night time to 33% and 32% at 24 and 12 m. Thus, there are clear differences in monoterpene species abundance between day and night.

These are mainly due to the nocturnal decreases in α-pinene and the nocturnal relative increase in limonene. It is plausible that the observed decrease in α-pinene mixing ratios could be due to a decreased vegetation emission, as reduced chemical destruction due to very low OH concentrations at night, would lead to an increase in the nocturnal α-pinene mixing ratios.

Even though there are clear differences between the absolute and relative abundances of some monoterpene species during day and night, there are no clear changes in the in the vertical gradients (e.g. for α-pinene night time averages were 0.15±0.05 for 12 m and 0.11±0.06 at 24 m). For the day, the apparent difference in the abundance of α-pinene is due to a single outlier data point covering 30 minutes at noon on 19 October 2015 at 24 m, when the α-pinene mixing ratio doubles. This increase could not be explained, although it could be related to a strong change in wind speed an hour before the measurement, when the wind was blowing from the North. In general, our observations indicate that the abundance of monoterpene species does not vary much over the heights selected (12 and 24 m) within the canopy. This is consistent with the results by Kesselmeier et al. (2000), where the monoterpene composition at the rain forest floor was comparable to the above-canopy composition at their site.




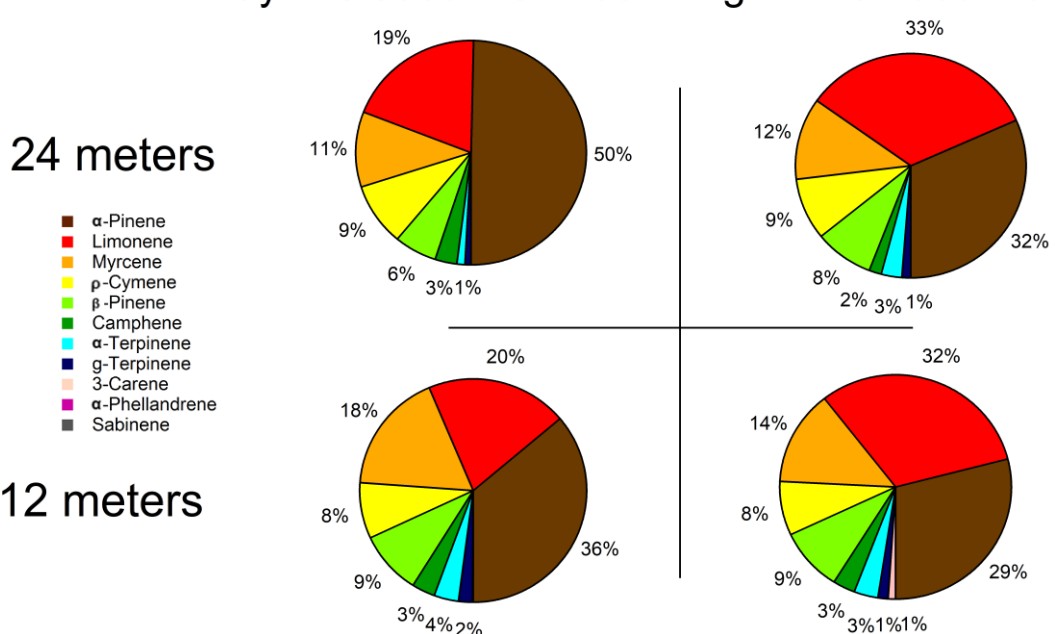

Figure 3: Pie charts representing day and night monoterpene species abundance.

3.3. Reactivity

The variability of the oxidants (OH, $O_3$ and $NO_3$) present in the Amazon air is important when

considering the impact that monoterpenes can have in the oxidative regime in the Amazon region and
Brazil. Hydroxyl radicals are produced mainly during the day via ozone photolysis and $O^1D$ reaction
with water. Low levels of OH can be also generated by the reaction of ozone with doubly bonded spe-
cies (e.g. monoterpenes and sesquiterpenes) even at night. In this assessment, we consider the monoter-
pene contributions to OH reactivity by day only. In contrast, $NO_3$ is photolytically destroyed during the
day, but can become significant at night so we assess the impact of monoterpenes on $NO_3$ reactivity at
night. Even though in the Amazon rainforest ozone levels are low (~10-20 ppb) compared to other areas
of the world (e.g. Williams et al., 2016), it is nevertheless present, and some monoterpenes are ex-
tremely reactive towards ozone. Table 2 gives an overview of the lifetime and reactivity to 1 ppb of all
the investigated monoterpene species for these three oxidants. For this Table 2, typical oxidant concen-
tration for the Amazon rainforest conditions were used. For OH a mean value of $7 \times 10^5$ molecules $cm^{-3}$





was used as representative of the site (Spivakovsky et al., 2000). During the measurement period the
observed average ozone value was 12 ppb at 24 m (Andreae et al., 2015), whereas the NO$_3$ mixing ra-
tios were taken from the MLC-CHEM model simulations that predicted mixing ratios of ~0.4 ppt.

Table 2: Lifetime of the different monoterpene species related to OH, O3 and NO3 for the OH daytime conditions at 24 m
and at 12 m. In addition, the reactivity to 1 ppb of the different monoterpene species is calculated.

| Monoterpenes investigated | Formula | Lifetime (minutes) | | | Reactivity to 1 ppb s$^{-1}$ | | |
|---|---|---|---|---|---|---|---|
| | | OH | O$_3$ | NO$_3$ | OH | O$_3$ | NO$_3$ |
| α-Pinene | C$_{10}$H$_{16}$ | 449 | 615 | 250 | 1.42 | 2.3E-06 | 0.17 |
| Camphene | C$_{10}$H$_{16}$ | 447 | 57422 | 2461 | 1.43 | 2.4E-08 | 0.02 |
| Sabinene | C$_{10}$H$_{16}$ | 400 | 623 | 155 | 1.60 | 2.2E-06 | 0.27 |
| β-Pinene | C$_{10}$H$_{16}$ | 320 | 3445 | 618 | 2.00 | 4.0E-07 | 0.07 |
| Myrcene | C$_{10}$H$_{16}$ | 71 | 110 | 141 | 8.98 | 1.3E-05 | 0.30 |
| α-Phellandrene | C$_{10}$H$_{16}$ | 132 | 17 | 21 | 4.84 | 8.1E-05 | 1.96 |
| Δ3-Carene | C$_{10}$H$_{16}$ | 271 | 1397 | 170 | 2.37 | 9.9E-07 | 0.24 |
| α-Terpinene | C$_{10}$H$_{16}$ | 103 | 2 | 11 | 6.24 | 5.6E-04 | 3.76 |
| ρ-Cymene | C$_{10}$H$_{14}$ | 1577 | >90000 | >90000 | 0.41 | 1.3E-09 | 2.7E-05 |
| Limonene | C$_{10}$H$_{16}$ | 145 | 246 | 127 | 4.41 | 5.6E-06 | 0.33 |
| γ-Terpinene | C$_{10}$H$_{16}$ | 140 | 369 | 53 | 4.57 | 3.8E-06 | 0.78 |
| Isoprene | C$_5$H$_8$ | 238 | 4069 | 238 | 2.69 | 3.4E-07 | 0.02 |


The most abundant species are α-pinene, limonene and myrcene. However, with respect to their
reactivities towards the different oxidants, their relative contribution to total monoterpene reactivity dra-
matically changes and α-pinene is not the dominant species anymore. For instance, α-terpinene domi-
nates ozone reactivity associated with monoterpene abundance both during the day and night, as well as
the nocturnal nitrate reactivity, despite the low mixing ratios measured for this compound.



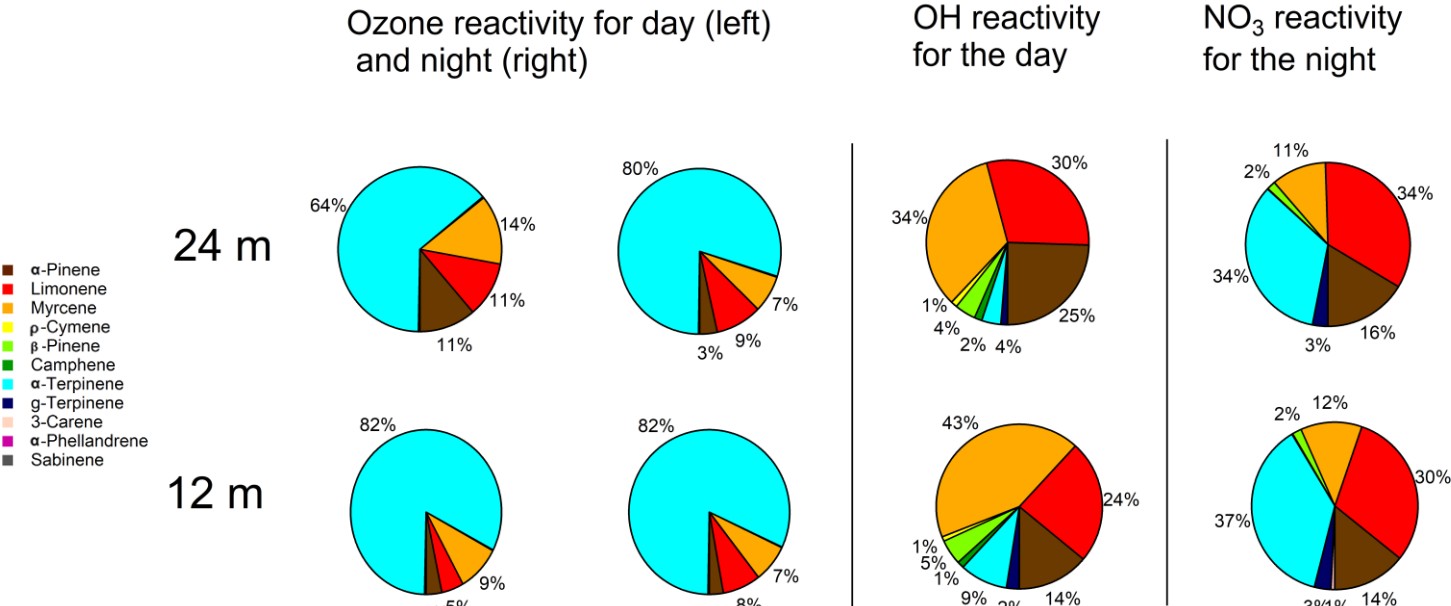

Figure 4: Pie charts representing day and night ozone reactivity, and OH reactivity (only for day) and NO$_3$ reactivity (only
for night), for 12 m on the bottom and 24 m on the top.

The monoterpene ozone reactivity seems to be quite similar between day (1.37x10$^{-06}$ s$^{-1}$) and night
(1.12x10$^{-06}$ s$^{-1}$). α-Terpinene dominates the monoterpene-ozone chemistry, followed by myrcene and
limonene. Despite the relatively high abundance of α-pinene (50%, average mixing ratio during the day
was 0.34±0.04 ppb at 12 m), its contribution to ozone reactivity with respect to other monoterpene spe-
cies was only 11% and 3% during the day and night, respectively at 24 m, and 3% for both day and
night at 12 m. As previously noted, the differences between heights are negligible for the night and
slightly higher at 24 m during the day. As ozone mixing ratios are quite similar for both heights during
day and night (11.4 ppb at 12 m and 10.4 ppb at 24 m during night, and 16.1 ppb at 12 m and 15.6 at 24
m during the day), the higher abundance of α-pinene during the day, and the lower α-terpinene mixing
ratios at 24 m during the day mainly explain these changes in monoterpene-ozone reactivity. It is im-
portant to note that these results are derived from a relative abundance analysis, and unmeasured mono-
terpene species could change the proportions, although given the close similitude between PTR-MS and
GC-FID measurements shown in Figure 1 this is unlikely. On the other hand, very reactive species





which could dominate reactivity, may be present in very low concentrations, and our measurements capabilities would not allow for its monitoring.

The monoterpene reactivity towards $NO_3$ radical during the night in this study is also dominated by α-terpinene (34 and 37%, respectively for 24 and 12 m), although contributions of limonene (30 and 34%, respectively for 24 and 12 m), α-pinene (16 and 14%, respectively for 24 and 12 m), and myrcene (11 and 12%, respectively for 24 and 12 m) are also significant. There seem to be no significant differences between the reactivity at different heights, suggesting a rather homogeneous chemical regime regarding monoterpene chemical destruction within the canopy (from 12 to 24 m). However, note that this finding reflects the use of a single simulated $NO_3$ concentration due to the absence of direct measurements in the Amazon rainforest, which prevents us from drawing any further conclusion. The OH reactivity estimates demonstrate the important role of myrcene with its higher reactivity towards OH due to its acyclic nature, especially at 12 m where myrcene is more abundant. The total OH reactivity for the sum of monoterpenes was calculated to be 2.3 and 3.4 $s^{-1}$ for 24 and 12 m, respectively.

### 3.4. Seasonality

By examining GC-FID data collected in previous campaigns, an intra- and inter-annual comparison can be made. These earlier samples were collected using a GSA SG10-2 personal pump sampler. Adsorbent tubes were filled at 167 $cm^3$ $min^{-1}$ (STP) air flow for 20 min. Total monoterpene averages for each season were calculated from 1100 to 1600 LT at 24 m. Based on these data, we distinguished the monoterpene mixing ratios representative for the dry season, the wet season and the wet-to-dry transition. The dry season conditions were represented by measurements collected in November 2012, September 2013, August 2014, and the measurements from this study in October 2015. The wet season measurements were collected in March 2013 and the wet-to-dry transition measurements were collected in June 2013. For the dry season conditions, the total monoterpene mixing ratios were substantially higher (1.02 ppb) compared to the observed monoterpene mixing ratios in the wet season (0.14 ppb) and





the wet-to-dry transition season (0.18 ppb) (Figure 5). This coincides with the occurrence of the highest
radiation levels and temperatures as well as the lowest precipitation during these dry season measure-
ment campaigns. During the wet season, the total monoterpene mixing ratios are lowest, while during
the transition season in June, they are slightly higher.

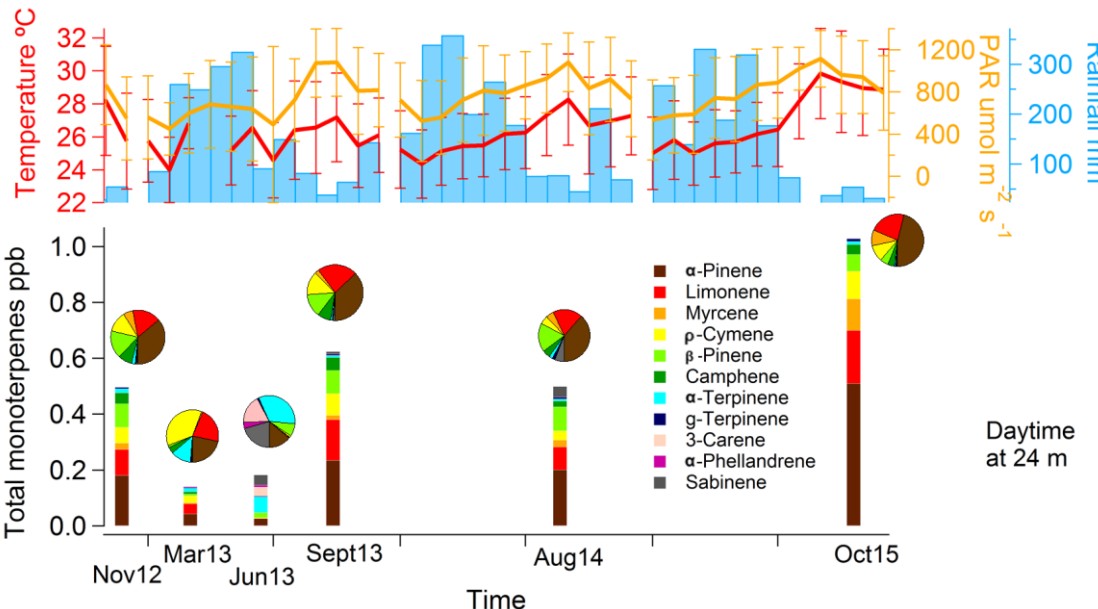

Figure 5: Graph showing the monoterpene speciation during the seasons of measurements. In the top, the monthly average of
temperature (in red) and photosynthetically active radiation (in orange) are displayed with their standard deviations for the
80 m height. The rain, also on top, is displayed as mm per month (bars). In the bottom the different monoterpene species are
differentiated by colours, stacked together adding up to the sum of monoterpenes. On top of each bar, a chart pie with the
chemical speciation is shown for easier visualization.

For each season, an average monoterpene chemodiversity distribution is shown in Fig. 5. During

the dry seasons the chemodiversity seems relatively similar (39.4±4% for α-pinene, 20.3±3% for limo-
nene), whereas it slightly changes during the wet season, and dramatically changes during the wet-to-
dry transition. The reason for this difference in June could be related to changes in the phenology, as
demonstrated at a Central Amazonian site (Alves et al., 2016; Lopes et al., 2016). Furthermore, during
the dry season of 2015 a very strong *El-Niño* event was taking place, leading to extremely dry condi-
tions observed region-wide (Jardine et al., 2017).



### 3.5. Modelling with MLC-CHEM

To further support our analysis of the observed magnitude as well as temporal variability in the monoterpene mixing ratio inside the forest canopy, we used MLC-CHEM to: 1) explore how well the model represents the measured mixing ratios and 2) to assess the role of the different in-canopy processes in explaining the diel cycle of the observed monoterpene mixing ratios at the ATTO site. We used the observed meteorological parameters and above-canopy $O_3$ concentration from the ATTO site during the period of measurement to constrain these simulations with MLC-CHEM. Meteorological observations for 18 October were missing and therefore, for this day the MLC-CHEM was driven by first-order estimates of the diurnal cycles in radiation, air and surface temperatures, relative humidity and wind speed based on the previous day's meteorological information. These simulations represent a set-up of MLC-CHEM distinguishing six canopy levels with a canopy height of 30 m, implying canopy layers with a thickness of 5 m. Furthermore, we assumed a Leaf area index (LAI) of 5 $m^2$ $m^{-2}$ and a Leaf Area Density (LAD) profile such that about 70% of this biomass is present in the top 15 m of the canopy. Monoterpene emissions by vegetation were simulated using a temperature-only dependent emission as a function of the amount of biomass in each layer and the measured canopy temperature profiles interpolating between the 0.4 m and 26 m temperature sensors.





Figure 6: Graphs showing the simulated results (solid lines) for 12.5 m (orange) and 22.5 m (green) from the MLC-CHEM, with the GC-FID speciated mixing ratios measurements for the most important monoterpene species at ATTO. The error bars represent the 20% uncertainty involved in the GC-FID measurements.

From Fig. 6, which shows a comparison of the simulated (12.5 and 22.5 m) and observed (12 and 24 m) speciated monoterpene mixing ratios from 17 to 20 October 2015, it can be inferred that the simulated speciated monoterpene mixing ratios are of comparable magnitude to the measured observations. Furthermore, it seems that overall the diurnal variability is well captured by the model. The processes involved in explaining the simulated temporal variability in the canopy monoterpene mixing ratios are: vegetation emission as a function of the basal emission factor and temperature dependency, leaf biomass and the canopy temperature measurements, turbulent mixing, chemical destruction and deposition to surfaces. Note that a temperature and light dependent emission algorithm was performed and the





modelling results did not follow the observed magnitudes and temporal variability as good as the modelling simulations using temperature only dependent emission. For instance, the comparison for the night of 17 to 18 October 2015 reflects the potentially important role of deposition to wet leaf surfaces. MLC-CHEM uses relative humidity as a proxy for the fraction of the leaf surface being wet. This seems to result in substantially smaller estimates of canopy wetness on 17 October 2015 compared to the following days, which explains also the simulated high nocturnal α-pinene mixing ratios, whereas the observations show actually minimum mixing ratios similar to those observed for the night of 18 to 19 October 2015. The simulated α-pinene mixing ratios for that night as well as for the remainder of the measurement period are in much better agreement with the observations. Regarding the comparison of the simulated observed mixing ratios for some of the other monoterpenes, the simulated ß-pinene, limonene, and myrcene mixing ratios, especially at 12.5 m seem to capture the observed temporal variability quite well. Note that this result for limonene reflects the use of a high leaf basal emission factor (2.25 µg C g$^{-1}$ hr$^{-1}$) required to simulate mixing ratios reaching up to 0.4 ppb. MLC-CHEM was applied to infer how much of the actual emission flux escapes the canopy expressed by the calculated atmosphere-biosphere limonene flux divided by the canopy emission flux of limonene. This ratio reaches a maximum value of 0.5 around noontime implying that these model simulations indicate that at the middle of the day, about 50% of the emitted limonene is removed inside the canopy by in-canopy chemistry and deposition. During night time, this ratio reaches a minimum $< 0.1$ indicating simulation of very efficient in-canopy removal.

## 4. Discussion

The observed differences in the monoterpene chemodiversity of the Amazon rainforest canopy atmosphere are driven by differences in emission, reactivity to the oxidizing species, physical removal processes and turbulent mixing conditions. As demonstrated in this data set, chemically speciated measurements are very important for understanding how monoterpenes affect Amazon air chemistry dependent on time of day and season. When comparing our results to previously published studies, we observe consistent differences with other regions of the Amazon rainforest. For instance, Kesselmeier et al.



(2002), studied the seasonal monoterpene speciation in the Rondonia rainforest. Even though they found the same monoterpene species as presented in this study, their individual abundances were very different compared to the mixing ratios for the dry season at the ATTO site. In the case of β-pinene, the abundance measured at ATTO was much lower than at other Amazonian sites (Andreae et al., 2002; Karl et al., 2007). These differences show that it cannot be assumed that the same speciation and emission rates of monoterpenes exist throughout the vast Amazon basin.

The emission of monoterpenes has generally been thought to be from storage glands in specialized structures like resin ducts, glandular trichomes or related structures (Schürmann et al., 1993; Steinbrecher, 1989). The release from these pools is governed by leaf temperature (Fall et al., 1999; Kesselmeier and Staudt, 1999; Monson et al., 1995), but other studies on monoterpene emissions have indicated that these emissions originate from recently photosynthetically fixed carbon (Staudt et al., 2017; Staudt and Bertin, 1998). In particular, Amazonian species have been found to show an emission dependency on light and temperature (Bracho-Nunez et al., 2013; Jardine et al., 2015; Kuhn et al., 2002, 2004). This could partly explain the diurnal pattern of α-pinene, which resembles the observed diurnal cycle in mixing ratios of VOCs such as isoprene with a light and temperature dependent emission flux (Kuhn et al., 2002; Rinne et al., 2002; Williams et al., 2007). However, this behaviour is not observed for all monoterpene species. Therefore, analysis of the observed diurnal cycles in some monoterpene species might lead to the conclusion, based on the assumption that this temporal variability is mainly determined by emissions and turbulent mixing ratios, that these emissions are both light and temperature dependent. Our analysis combining the observations and the canopy exchange modelling system, however indicates that this temporal variability seems to be well explained using temperature dependent emissions alone, combined with the role of the in-canopy chemical destruction as well as the potentially important role of canopy deposition, in part to wet leaf surfaces.

It has been shown previously that the amounts and speciation of monoterpenes vary strongly according to plant species and leaf developmental stage. For instance, Bracho-Nunez et al. (2011) found young leaves of the some Mediterranean plant species to emit more α-pinene and mature leaves to emit



*e*-ocimene, *z*-ocimene and myrcene, but not α-pinene. Some species have been found to be higher emit-
ters of α-pinene (i.e. *Hevea spruceana*), whereas others are higher emitters of myrcene (i.e. *Quercus*
*coccifera,* Bracho-Nunez et al., 2013). The leaf developmental stage is also important, as reported for
flushing young leaves emitting monoterpenes in contrast to the isoprene emission of mature leaves of
the same plant species (Kuhn et al., 2004). Such a behaviour could explain the lower mixing ratios and
different chemodiversity found in June. During this time of the year, leaf flushing takes place in the
Central Amazon region (Alves et al., 2016; Lopes et al., 2016). Under these conditions, lower α-pinene
mixing ratios were found as compared to the dry season, when young leaves reach mature levels. There-
fore, the seasonality in Amazon forest monoterpene emissions might depend more on the changes in ag-
gregated canopy phenology than on the seasonality of climate drivers (Wu et al., 2016). In this study we
have shown that chemodiversity remained relatively constant during at least the dry seasons, but
changed between different seasons, and that the implications to the atmosphere are different for each
monoterpene species. Kesselmeier et al. (2002) also showed this type of behaviour in their study, where
they did not find a strong difference in total mixing ratios, but different chemodiversity, between sea-
sons, likely expressing differences in seasonal plant developments and reactivities, which should be ac-
counted for model implementation at the ATTO site.
Another driver for the observed difference in chemodiversity in the Central Amazon rainforest
canopy is the difference in oxidation rates. Therefore, a lower abundance of a certain monoterpene spe-
cies could not only be related to a lower vegetation emission, but also to a higher reactivity with atmos-
pheric oxidant species. Despite the small amount of α-terpinene present in the atmosphere, it can pro-
foundly affect reactivity due to its  fast reaction rate (its lifetime, according to the oxidant mixing ratios
stated above, can be 103, 2 and 11 minutes to OH, $O_3$ and $NO_3$, respectively (Neeb et al., 1997). In
terms of total OH reactivity accounted for by the MTs, the values of this study are very low compared
to the total OH reactivity measurements by Nölscher et al. (2016), with a mean of total OH reactivity
for the dry season of 32 $s^{-1}$, mostly dominated by isoprene chemistry. This suggests that the monoter-
penes contribute only a small fraction to the total OH reactivity at the ATTO site. This study demon-
strates that the abundance does not relate to the importance in chemical reactivity, and species that are



usually not considered into atmospheric chemistry models due to their modest mixing ratios might actu-
ally play a dominant role in the monoterpene atmospheric chemistry. Therefore, it is not correct to gen-
eralize the representation of terpene chemistry in models (Hallquist et al., 2009) using one or two mono-
terpene species only.

The gas-phase oxidation of the monoterpenes in the Amazon has numerous impacts on the envi-
ronment including the production of a multitude of new compounds that are generally longer lived than
the primary emissions, increasing the lifetimes and particle production potential of certain compounds
by suppressing oxidant availability. Moreover, production of OH due to the ozonolysis of monoterpenes
is known to occur (Paulson et al., 1999). The production strength varies depending on the position of
the double bonds (if there is more than one) (Herrmann et al., 2010). Furthermore, the products of the
reaction can be manifold. For instance, when α-pinene is oxidized by OH, especially at low nitrogen ox-
ides mixing ratios, pinonaldehyde is formed in high yields (Eddingsaas et al., 2012). Chemical pro-
cessing of α-pinene can also result in a further production of different monoterpenes such as the reaction
of α-pinene with nitrate during the night, which can lead to the formation of ρ-cymene (Gratien et al.,
2011).


The implications of the measured monoterpene abundances for SOA formation at the ATTO site
are difficult to quantify. For an aerosol to be formed certain conditions must be met, as the aerosol yield
of the parent biogenic hydrocarbon depends on the concentration of organic aerosol into which these
products can be absorbed (Griffin et al., 1999). This is the reason why in regions with similar monoter-
pene mixing ratios and different aerosol loading, the SOA yield can vary. For instance, α-pinene forms
no aerosol under $NO_3$ oxidation (Fry et al., 2014), whereas there is production of aerosols when the oxi-
dation of α-pinene involves $O_3$ (Ehn et al., 2014) and OH (Eddingsaas et al., 2012). Monoterpenes con-
taining endocyclic double bonds (e.g. α-pinene, 3-Carene) or open chains (e.g. myrcene) tend to form
less aerosols from ozonolysis than monoterpenes with exocyclic double bonds (e.g. β-pinene, sabinene,
Hatakeyama et al., 1989; Hoffmann et al., 1997). Following the equation established by Bonn et al.
(2014), we were able to quantify the potential aerosol growth from the monoterpene species alone



$(5\times10^{-6}$ to $5\times10^{-5}$ molec cm$^{-3}$ at 24 m) being two orders of magnitudes less than the potential aerosol
production from sesquiterpenes, assuming mixing ratios of the latter of 0.2 ppb $(5.21\times10^{-4}$ and $3.43\times10^{-3}$
molec cm$^{-3}$ at 24 m) based on previous measurements in the Amazon (Jardine et al., 2011). Further-
more, the level of NO (nitric oxide) present also affects severely the potential aerosol growth, for in-
stance in the atmosphere a change from 0.2 ppb to 1 ppb of NO leads to a decrease in the formation rate.
This interdependence calls for a consistent consideration of the BVOC and NO$_x$ exchange in aerosol
formation and growth studies. For both species, consideration of the demonstrated canopy interactions
are very important, not only in terms of determining the overall average flux out of the canopy, but also
in terms of the temporal variability in atmosphere-biosphere fluxes.

The modelling results have indicated gaps in the understanding of the common processes affect-
ing the monoterpene species dynamics. Based on the observations from this study, deposition to wet
surfaces may play an important role in the removal of monoterpene species with different reactivities
and solubility. This potentially important role of canopy deposition could have two important implica-
tions: 1) the effective emissions into the atmosphere are substantially smaller compared to leaf-scale
based emission flux estimates and 2) monoterpene emission flux estimates based on above-canopy mix-
ing ratio measurements can be substantially smaller than the actual leaf-scale monoterpene emission
flux. These findings should be considered in the further development of inventories for application in
large/scale chemistry models.

5. Conclusions
This study presents an analysis of the measured monoterpene chemodiversity at the Amazon
tropical forest measurement site, ATTO. The results showed a distinctly different chemical speciation
between day and night, whereas there were little vertical differences in speciation within the canopy (12
and 24 m). Furthermore, inter- and intra-annual results demonstrate similar chemodiversity during the
dry seasons analysed, but this chemodiversity changed with season, similar to the seasonal measure-
ments performed by Kesselmeier et al. (2002). Furthermore, reactivity calculations demonstrated that

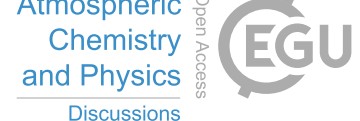

the most abundant compounds may not be the most atmospheric chemically relevant compound, and therefore it is not correct to generalize the representation of terpene chemistry in models. Moreover, simulations with a canopy exchange modelling system to assess the role of canopy interactions compared relatively well with the observed temporal variability in speciated monoterpenes, but also indicate the necessity of more experiments to enhance our understanding of in-canopy sinks of these monoterpenes.

## 6. Data Availability

Even though the data are still not available in any public repository, the data are available upon request from the main author.

## 7. Acknowledgements

The authors thank the Max Planck Society and the Instituto Nacional de Pesquisas da Amazonia for continuous support. Furthermore, we acknowledge the support by the ATTO project (German Federal Ministry of Education and Research, BMBF funds 01LB1001A; Brazilian Ministério da Ciência, Tecnologia e Inovação FINEP/MCTI contract 01.11.01248.00), UEA and FAPEAM, LBA/INPA, and SDS/CEUC/RDS-Uatumã. We would especially like to thank all the people involved in the logistical support of the ATTO project, in particular Reiner Ditz and Hermes Braga Xavier. We acknowledge the micrometeorological group of INPA/LBA for their collaboration concerning the meteorological parameters, with special thanks to Antonio Huxley and Leonardo Oliveira. We are grateful to Nina Knothe for logistical help. We greatly acknowledge Guenther Schebeske for the GC-FID analysis. We would also like to thank Thomas Klüpfel, Tomas Chor and Emilio Hoeltgebaum for their help during sampling. This paper contains results of research conducted under the Technical/Scientific Cooperation Agreement between the National Institute for Amazonian Research, the State University of Amazonas, and the Max-Planck-Gesellschaft e.V.; the opinions expressed are the entire responsibility of the authors and not of the participating institutions.

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
