# Peer review of "Atmospheric Chemistry and Physics Discussions"

_Atmospheric Chemistry and Physics, 2017_

## Referee Comment (RC1) · Anonymous Referee #1 · 27 Oct 2017

General:

The paper presents in-canopy, speciated BVOC measurements by GC-FID over a period of three days. These are compared PTR measurements and to longer time series of GC measurements in different seasons at the same site. The data is analyzed and discussed in terms of emission height, diurnal cycle, reactivity, and seasonality in comparison to the longer data series in different seasons. Observations are compared to a canopy model which contained detailed chemistry for some species and rudimentary chemistry for new others. The purpose of the comparison is to demonstrate that

speciated measurements are a good test if the overall key features and processes are captured by the model. The manuscript presents interesting data and data analysis. The text is well structured. The manuscript could be published in ACP after considering the following major and minor comments.

My major concern relates to the discussion of the role of deposition on wet leaf surfaces. From the presented material I do not see proof - from the model comparison- that deposition on wet leafs is a significant process. This may be in parts owed to the fact that description and interpretation are not detailed enough. The authors may address the following questions and issues.

(I think a paragraph should be started in line 406 where the leaf wetness is discussed.)

1. Did you observe the leaf surface wetness in the two nights? If yes, was it the same or was it different?

2. You suggest that the model has a strong missing sink in the night from the 17. to the 18. Is this the only possible explanation for the strongly overpredicted a-pinene mixing ratio? Or could be there more reasons for the model showing so high a-pinene in the night from the 17. to the 18.? Only if the source and the chemical sink of a-pinene are about the same in the two nights then the depositions sink must have been also similar as the mixing ratios are about the same. Were source and chemical sink the same in both nights?

3. How can deposition on wet leaves affect so strongly the mixing ratio of non-water soluble compounds like a-pinene?

4. I understand there was no RH data for the 18.. How could the model derive a reliable surface wetness then?

5. Why do the different MT respond so differently to the ill predicted surface wetness on 17.: a-pinene and b-pinene too high, limonene ok, myrcene in between, and terpinolene too low. Insofar I think the statement in line 527 is not justified by the presented

material and analysis.

Major comments:

line 514-518: I do not understand what the authors are discussing here. I cannot understand how a "potential" aerosol growth could have a dimension of molecules per cm3 and an order of 10-5 -10-6. Moreover, I did not find such notations in Bonn et al. 2014. This part must be clarified or deleted.

line 519f: This a strong statement. Do you have proof for that or a reference? Moreover, potential aerosol growth and formation rate are not the same. Indeed NO suppresses nucleation (Wildt et al. ACP, 2014) but not so much the yield (Sarrafzadeh et al. ACP, 2016)

Minor comments:

Starting with line 219 the authors give errors/uncertainty ranges/standard deviations in form of "+/-yx" at many instances (including Table 1) without explaining the specific meaning.

line 156: "142 Td", Td is not explained

line 186f: The formulation "selected" suggests to me that the initial EA and the fractional EA for individual compounds were used as adaptable parameters. Is this case? If so, then the overall good agreement between modelled and observed concentrations in section 3.5 (line 398f) and Figure 6 is not surprising. If not, explain the in more detail the rationale for these selections.

line 386f: Similar is true for the formulation about the "assumed" lead area index and leaf area density. What is the rationale for these choices? Did you estimate it from observations on the site?

line 256, first §: These percentages are averages over day and night hours and the three measurement days? They should have standard deviations.

line 286, first §: I suggest to define what you mean by reactivity. e.g. OH reactivity = kOH*[MT].

line 303, Table 2: The header of the third column is misleading. It should read "normalized reactivity for 1 ppb [s-1]", or so. Otherwise confusion with the use of reactivity later in the text.

line 489: Is Hallquist 2009 (a review) a good reference for this statement?

line 542f: That is not new and with the given formulation the conclusion in its generality does not make sense. The authors did not show that there is no suited representation of MT, which considers also their reactivity. Moreover, the degree of tolerable simplification depends also on the purpose of the model calculation.

---

## Referee Comment (RC2) · Anonymous Referee #2 · 14 Nov 2017

General comments:

The manuscript "Monoterpene chemical speciation in the Amazon tropical rainforest: variation with season, height, and time of day at the Amazon Tall Tower Observatory (ATTO)" is a suitable and in the scope of "Atmosphere Chemistry and Physics", shading light on the importance of monoterpene characterisation. Although, the manuscript is interesting it requires a major revision as it lacks the clear structure. The reader is often forced to go forward and beck. Many discussions are found in Results (e.g. L230-232, L241-242, L267-270, L276-278, L280-281 etc.). Many statements are started

with not suitable paragraph context (e.g. L306: "The most abundant species are $\alpha$-pinene, limonene and myrcene".). Some statements and discussion are out of context or even incorrect as mentioned in the Specific comments below. It is also too long for the scientific content. I suggest the following: Restructure the text in order to join Results and Discussion into one section with adequate subsections, redo the figures as suggested below and delete the repetitions in the text. Please also find specific and technical comments.

Specific comments:

L60-63. Not entirely true. PTR-MS could have a time resolution <1s (not 30 s). It is also recently coupled with FastGC to characterise monoterpenes in < 2min. 1h GC-FID is a bit too much for "current" method. Please update the references for all this. L103 and 125 – Not indicated sampling frequency, 30 min mentioned in L125, is not in agreement with the resolution presented in the Figure 1b and c. A detailed but structured description of the methods used is needed. L106 – Air sampling section needs structure improvement. You first open with your sampler giving the reference, then you describe your sampling dates/times and then you come back to the sampler, and again back to the sampling procedure. L224 – In Table 1 it is not clear what the tolerance is. L128 – The "Instrument for chemical analysis". Are you intent to describe the instruments or the methods used? L137 – What is "rapidly" in this context? L160 – Why just 12 and 24 if all the data are available? L164 – "$2\sigma$ of the background" – $3\sigma$ it is more acceptable. Also briefly describe the blanks in each sampling systems. L239 - Isoprene! Not mentions in material and methods. How is this measured? L253 – Figure not clear. E.g. isoprene 24m not visible after 12:00. Legend not descriptive, not explained what are the error bars. Some error bars below zero and yet, above LOD? Explain? L345 – "..in previous campaigns.. earlier samples were collected using a GSA SG10-2 personal pump sampler. Adsorbent tubes were filled at 167 cm3 min-1 (STP) air flow for 20 min". This is again an example of poor structure. This need to be in the Material and Methods section. L380 - How the above-canopy O3 concentrations used in the model

and for the reactivity calculation represent real situation in the canopy (between 12 and 24m)? L439 and on – "The emission of monoterpenes has generally been thought to be from storage glands in specialized structures like resin ducts, glandular trichomes or related structures (Schürmann et al., 1993; Steinbrecher, 1989)." This generalization and discussion based further in the text are related only to conifer type or monoterpene emitters (see also in your references). Thus, this is irrelevant (and incomparable) for the tropical forest as the physiology (and chemotypes) for monoterpene emission in broadleaf tree species is different to the pool emitters. L502 and 522 – Any leaf level experiments to address this? Needs a brief discussion.

Technical corrections:

L1 – Two words "Amazon" in a title are not needed. L19 – Why just in Amazon rainforest? You may just say "a rainforest". L24 – "automatic" - automated? L31 – "may not be the most atmospheric chemically relevant compounds". Although it might be grammatically correct, this is a bit odd and does not read well. This is repeated later in the text. L74-76 – "scarce" – a bit over-repetition of this word throughout the manuscript and here. L230 – The is a discussion in the "Result" section. L253 – Figure 2 legend and axes text too small. L283 – Figure 3 – figure caption above not needed. Description needs to be extended. L289 – "O1D" ?! L312 – Figure 4 out of margins, figure caption not needed. Use a) b)... to refer to the individual figures with a clear description. Also, check all the figures to meet this standard. L381-384 - Repetition (missing meteorological information and how the i

---

## Author Comment (AC1) · 22 Jan 2018

1- Comments from reviewers 2- Author's response 3- Change in the text

Reviewer #1 (Remarks to the Author):

General: The paper presents in-canopy, speciated BVOC measurements by GC-FID over a period of three days. These are compared PTR measurements and to longer time series of GC measurements in different seasons at the same site. The data is analyzed and discussed in terms of emission height, diurnal cycle, reactivity, and

seasonality in comparison to the longer data series in different seasons. Observations are compared to a canopy model which contained detailed chemistry for some species and rudimentary chemistry for new others. The purpose of the comparison is to demonstrate speciated measurements are a good test if the overall key features and processes are captured by the model. The manuscript presents interesting data and data analysis. The text is well structured. The manuscript could be published in ACP after considering the following major and minor comments.

1- My major concern relates to the discussion of the role of deposition on wet leaf surfaces. From the presented material I do not see proof - from the model comparison that deposition on wet leafs is a significant process. This may be in parts owed to the fact that description and interpretation are not detailed enough. The authors may address the following questions and issues.

2- We understand the concern of the reviewer. We use a canopy exchange model which has been previously extensively evaluated for different ecosystems including tropical rainforest. We have further stressed this feature of model evaluation and have included some further references for more detailed information about some of the canopy model features relevant to the presented subject of BVOC exchange, e.g., estimating canopy wetness as a function of RH. We are not claiming that deposition to leaf wet surfaces is a significant process. One would actually indeed expect this sink to be of minor relevance for the terpenes included in this study given their low solubility's. However, the inferred wet vegetation uptake resistance following the widely-used Wesely (1989) approach of $\sim$300 s m-1 results in simulation of an apparent not ignorable sink also given the large area of potential wet surface. Using this inferred uptake rate we achieved a relative good agreement between model simulated and observed monoterpene mixing ratios, not only in magnitude (which is no surprise given the fact that the constant basal emission factor has been selected to reproduce the campaign average mixing ratios) but especially regarding the diurnal dynamics. The results point to a combined effect of the potential role of chemistry and deposition as monoterpene

sinks pointing to the necessity to explore further this deposition to wet surface process.

1- (I think a paragraph should be started in line 406 where the leaf wetness is discussed.)

2- We have separated a paragraph for sink processes.

1. Did you observe the leaf surface wetness in the two nights? If yes, was it the same or was it different?

2- Unfortunately we did not have measurements of leaf surface wetness available for the site, and therefore we used relative humidity as a proxy for the calculation of leaf surface wetness. MLC-CHEM uses relative humidity as a proxy for the fraction of the wet leaf surface (Lammel, 1999). This results in smaller estimates of canopy wetness on 17 October 2015 compared to the following days. In the revised version, we now discuss that these estimates of canopy wetness cannot be corroborated without measured canopy wetness.

1. You suggest that the model has a strong missing sink in the night from the 17. to the 18. Is this the only possible explanation for the strongly overpredicted a-pinene mixing ratio? Or could be there more reasons for the model showing so high a-pinene in the night from the 17. to the 18.? Only if the source and the chemical sink of a-pinene are about the same in the two nights then the depositions sink must have been also similar as the mixing ratios are about the same. Were source and chemical sink the same in both nights?

2. In response to this comment we modified the discussion about the comparison of the simulated and observed temporal variability in monoterpene mixing ratios and the role of canopy wetness. Rather than stressing these contrasts between the night of 17-18 October and the other nights we now contrast more generally the 17th and the other three days. This approach was chosen due to missing meteorological observations for the 18th. There is a significant change in the modeled canopy wetness conditions

around midnight due to using a prescribed typical diurnal cycle in some meteorological parameters instead of the actual observations for the 17th. It provides though an interesting sensitivity analysis since as soon as the imposed canopy wetness increases to a value of 1, $\alpha$-pinene drops to much lower values. This strong dependence of the simulated temporal variability on changes in canopy wetness conditions is confirmed by an analysis of the process tendencies in the model which show that the emissions do not significantly change comparing the 17th with the other 3 days of the campaign. The contribution by chemical oxidation also shows some temporal variability due to differences in O3, OH and NO3 but this sink is relatively small compared to the dominant sink of $\alpha$-pinene, canopy deposition. There the changes in canopy wetness contrasting the 17th with the other 3 days explains to a large extent the simulated temporal variability in mixing ratios of $\alpha$-pinene and the other terpenes and which generally agrees quite well with the observed temporal variability.

1- How can deposition on wet leaves affect so strongly the mixing ratio of non-water soluble compounds like a-pinene?

2- What we present here, relies on application of inferred uptake efficiencies for these monoterpenes not only commonly being used in any model study following the Wesley (1989) approach to consider dry deposition but also recently being applied in a detailed study on Boreal forest canopy exchange of BVOCs (Zhou et al., 2017). The inferred monoterpene wet-surface uptake resistances on the order of 300 s m-1, based on Henry law's constants on the order of ~1-3e-2 M atm-1, suggest a potentially still quite efficient uptake by leaf wetness. Note that this seems to be similar to the observed removal of ozone by wet canopies where, despite its low solubility, there is actual experimental evidence that canopy wetness enhances O3 deposition potentially due to aqueous-phase chemical interactions. Clearly, this feature on leaf wetness in monoterpene removal has to be corroborated by further experimental evidence.

1- I understand there was no RH data for the 18. How could the model derive a reliable surface wetness then?

2- See also our response to the previous points about missing canopy wetness observations and the impact of the imposed meteorological data for the 18th of October.

1- Why do the different MT respond so differently to the ill predicted surface wetness on 17.: a-pinene and b-pinene too high, limonene ok, myrcene in between, and terpinolene too low. Insofar I think the statement in line 527 is not justified by the presented material and analysis.

2- In the revised version we have excluded the respective statement.

Major comments:

1- line 514-518: I do not understand what the authors are discussing here. I cannot understand how a "potential" aerosol growth could have a dimension of molecules per cm3 and an order of 10-5 -10-6. Moreover, I did not find such notations in Bonn et al. 2014. This part must be clarified or deleted.

2- Our aim was to discuss the effects of mono- and sesquiterpenes as classified in the Amazon on particle formation rates i.e. not aerosol growth rates. Admiringly, the reviewer is correct as the wrong wording was used. The units provided are wrong as the unit needs to be "cm-3 s-1".These have been corrected in the revision. A particle growth would focus on size change per time i.e. nm h-1 for a certain initial size but not change concentrations per time as provided. In general, we used the word "potential" as no direct measurements of particle growth or of particle size distribution were available and calculations are based on the transfer of results from elsewhere to Brazil. With respect to the aerosol formation and growth rate, it has been shown (for example by Wolf et al. (2009; 2011) and by Hummel (2010) and included in Bonn et al. (2014)) that the initial growth of (smaller) particles is determined by organic peroxy radicals as one of two processes. Those processes are (a) the reactive interaction of large organic peroxy radicals (RO2) from e.g. mono- and sesquiterpene reactions with OH or NO3 and (b) in case sufficient organic mass has been acquired already uptake by partitioning of oxidized VOCs with a reduced volatility. Both processes can be simulated and

large organic RO2 concentrations approximated assuming steady-state conditions for radicals which is a common assumption. For clarification the text has been changed to:

3- L388 (please refer to supplement for correct location of line number): "Following the equation established by Bonn et al. (2014) (Equation number 5 in text), we were able to estimate the potential aerosol particle number formation rate initiated by monoterpene species only ($1 \times 10^{-5}$ to $5 \times 10^{-5}$ cm$^{-3}$ s$^{-1}$ at 24 m) assuming steady state conditions for radicals. Those were found to be approximately two orders of magnitude smaller than the calculated potential new aerosol particle formation rate caused by oxidation products of sesquiterpenes. Our calculations assume mixing ratios of sesquiterpenes of 0.2 ppb revealing potential formation rates of $1 \times 10^{-3}$ and $4.5 \times 10^{-3}$ cm$^{-3}$ s$^{-1}$ at 24 m) based on previous measurements in the Amazon (Jardine et al., 2011) which are remarkably smaller than observed at mid-latitude conditions (Bonn et al., 2014)."

1- line 519f: This a strong statement. Do you have proof for that or a reference? Moreover, potential aerosol growth and formation rate are not the same. Indeed NO suppresses nucleation (Wildt et al. ACP, 2014) but not so much the yield (Sarrafzadeh et al. ACP, 2016)

2- We kindly refer the reviewer to Sarrafzadeh et al. (2006) and the respective Figure 1 that illustrates how SOA yield depends on the BVOC/NOx ratio (see attached figure 1). For Amazonian conditions (in contrast to smog chamber measurement conditions) the typical BVOC mixing ratios are smaller than 3 ppb and most of the time around 1 ppb. If we apply Fig. 1 of Sarrafzadeh et al. (2016), the situation is located on the very left of the plot. Increasing NOx at constant BVOC mixing ratio will decrease the BVOC/NOx ratio and lead to a decline in SOA yield. The unaffected region of the same figure is not applicable for the Amazon region because of present concentrations. This is in line with the findings of Wildt et al. (2014) for smaller particles indicating similar processes responsible for the growth. For clarification we provide with a more detailed explanation as below:

3- L396: "Furthermore, the level of present NO (nitric oxide) also affects the potential aerosol growth (Wildt et al., 2014) and yield (Sarrafzadeh et al., 2016) at low BVOC/NOx ratios. As the theory assumes contributions of larger organic peroxy radicals (RO2), which are destroyed by reactions e.g. with NO, increasing NOx at constant BVOC mixing ratio will decrease the BVOC/NOx ratio and lead to a decline in SOA yield. Our calculations showed this effect, with a change of NO from 0.2 ppbv to 1 ppbv leading to a decrease in the formation rate at a diameter of 3 nm. This interdependence calls for a consistent consideration of the BVOC and NOx exchange in aerosol formation and growth studies."

Minor comments:

1- Starting with line 219 the authors give errors/uncertainty ranges/standard deviations in form of "+/-yx" at many instances (including Table 1) without explaining the specific meaning.

2- We have added averages +/- standard deviation where corresponded in the text.

1- line 156: "142 Td", Td is not explained

2- This is now explained in the text.

1- line 186f: The formulation "selected" suggests to me that the initial EA and the fractional EA for individual compounds were used as adaptable parameters. Is this case? If so, then the overall good agreement between modelled and observed concentrations in section 3.5 (line 398f) and Figure 6 is not surprising. If not, explain the in more detail the rationale for these selections.

2- The emission factors from the selected monoterpenes were indeed adaptable parameters. We used selected constant basal emissions for terpenes for the plant ecosystem class, in this case tropical rainforest. We used initially basal emission factor for monoterpenes for tropical rainforest of 0.4 ug C g-1 hr-1 (Guenther et al., 1995). Then we partitioned this emission flux over the different monoterpene species to see
how well the model would reproduce the observed mixing ratios. Following, we adjusted the value for each species until the observed 4-day average mixing ratios were reasonably well reproduced. Given the daily changes in the processes potentially involved in explaining the observed temporal variability, the presented analysis gives you information about the role of the different processes, as these have different dominating effects at different times of the day and depend, at the same time, on many other drivers. In the text we have chosen to show the basal emissions in mg C g-1 h-1, rather than emission factor constant, and we have modified the text as follows in order to better explain this issue.

3- L179: "The monoterpene basal leaf-scale emission factors have been selected as such that the model simulates monoterpene mixing ratios of comparable magnitude compared to the campaign average observed mixing ratios. In the evaluation of simulated and observed mixing ratios we mainly focus on the comparison between the simulated and observed temporal variability being determined by the differences in canopy processes for contrasting nocturnal and daytime conditions. For the model simulation, the basal emission factors were 0.18 $\mu$g C g-1 h-1 for $\alpha$-pinene, 0.04 $\mu$g C g-1 h-1 for $\beta$-pinene, 0.11 $\mu$g C g-1 h-1 for $\alpha$-terpinene, 0.9 $\mu$g C g-1 h-1 for limonene and 0.18 $\mu$g C g-1 h-1 for myrcene. Note the selected relative high basal emission flux for limonene is required to reach simulated mixing ratios comparable to the observed ones."

1- line 386f: Similar is true for the formulation about the "assumed" lead area index and leaf area density. What is the rationale for these choices? Did you estimate it from observations on the site?

2- LAI was previously measured at the site, and we have included this reference in the methodology.

1- line 256, first §: These percentages are averages over day and night hours and three measurement days? They should have standard deviations.

2- We agree with the reviewer and have added the standard deviations of the percentages both in the graph and text.

1- line 286, first §: I suggest to define what you mean by reactivity. e.g. OH reactivity = kOH*[MT].

2- We have added this clarification in the text.

1- line 303, Table 2: The header of the third column is misleading. It should read "normalized reactivity for 1 ppb [s-1]", or so. Otherwise confusion with the use of reactivity later in the text.

2- We have modified this in Table 2 accordingly.

1- line 489: Is Hallquist 2009 (a review) a good reference for this statement?

2- We agree this is not the most appropriate reference and we have excluded it.

1- line 542f: That is not new and with the given formulation the conclusion in its generality does not make sense. The authors did not show that there is no suited representation of MT, which considers also their reactivity. Moreover, the degree of tolerable simplification depends also on the purpose of the model calculation.

2- We have reformulated the sentence, and we believe the point of the sentence was not well stated before.

1- L502 "Furthermore, reactivity calculations demonstrated that higher abundance of MT does not automatically imply higher reactivity as the most abundant compounds may not be the most atmospheric chemically relevant compound or the relative contribution of different monoterpenes may change. Our calculations support the view to that the role of canopy exchange may be erroneously estimated when not taking into account speciation based reactivity in models. Moreover, simulations with a canopy exchange modelling system to assess the role of canopy interactions compared relatively well with the observed temporal variability in speciated monoterpenes, but also indicate the necessity of more experiments to enhance our understanding of in-canopy

sinks of these monoterpenes."

Reviewer #2 (Remarks to the Author):

General comments:

1- The manuscript "Monoterpene chemical speciation in the Amazon tropical rainforest: variation with season, height, and time of day at the Amazon Tall Tower Observatory (ATTO)" is a suitable and in the scope of "Atmosphere Chemistry and Physics", shading light on the importance of monoterpene characterisation. Although, the manuscript is interesting it requires a major revision as it lacks the clear structure. The reader is often forced to go forward and beck. Many discussions are found in Results (e.g. L230-232, L241-242, L267-270, L276-278, L280-281 etc.). Many statements are started with not suitable paragraph context (e.g. L306: "The most abundant species are $\alpha$-pinene, limonene and myrcene".). Some statements and discussion are out of context or even incorrect as mentioned in the Specific comments below. It is also too long for the scientific content. I suggest the following: Restructure the text in order to join Results and Discussion into one section with adequate subsections, redo the figures as suggested below and delete the repetitions in the text. Please also find specific and technical comments.

2- We thank the reviewer for the fruitful feedback. As suggested, we have now merged the results and discussion sections, removed repetitive sentences and redid the figures.

Specific comments:

1- L60-63. Not entirely true. PTR-MS could have a time resolution <1s (not 30 s). It is also recently coupled with FastGC to characterise monoterpenes in < 2min. 1h GCFID is a bit too much for "current" method. Please update the references for all this.

2- We agree with the reviewer and therefore we have removed that part from the text.

1- L103 and 125 – Not indicated sampling frequency, 30 min mentioned in L125, is not in agreement with the resolution presented in the Figure 1b and c. A detailed but

structured description of the methods used is needed.

2- The sampling frequency was every hour for 30 minutes. We agree with the reviewers this is not clear in the text and we have modified this paragraph for better explanation.

3- L111 "The samples were collected from 17 to 20 October 2015 at an hourly frequency. Samples were collected for 30 min every hour at a flow of 200 cm3 min-1 (STP), leading to a collection of 6 L of air in each cartridge."

1- L106 – Air sampling section needs structure improvement. You first open with your sampler giving the reference, then you describe your sampling dates/times and then you come back to the sampler, and again back to the sampling procedure.

2- We thank the reviewer for this comment, as it is true that section needed improvements. Therefore, we have restructured this section following:

3- L97: "Collection of ambient air samples on adsorbent tubes, for subsequent analysis by Gas Chromatography – Flame Ionization Detector (GC-FID), was made with two automated cartridge samplers described earlier (Kesselmeier et al., 2002; Kuhn et al., 2002, 2005) positioned at 12 and 24 m on the INSTANT tower. The samplers consist of two main units, a cartridge magazine that holds the adsorbent-filled tubes and the control unit timing the process and recording the data. This latter unit also houses the pumps (Type N86KT, KNF Neuberger, Freiburg, Germany), pressure gauges, mass flow controllers and power supply. The cartridge magazine is equipped with solenoid valves controlling the inlet and outlet of up to 20 individual sampling adsorbent tubes. The system is a constant-flow device, with one cartridge position per loop used as a bypass for purging the system. Due to the compact weatherproof housings and the low power consumption, we were able to position one sampler at 24 m and the other one at 12 m, attached to the INSTANT tower booms with commercially available 50 mm aluminium clamps. The adsorbent tubes used for VOC sampling were filled with 130 mg of Carbograph 1 (90 m2 g-1) followed by 130 mg of Carbograph 5 (560 m2 g-1) sorbents. The size of the Carbograph particles was in the range of 20–40 mesh.

[Figure]

Carbographs 1 and 5 were provided by L.A.R.A s.r.l. (Rome, Italy) (Kesselmeier et al., 2002). The samples were collected from 17 to 20 October 2015. Samples were taken for 30 min every hour at a flow of 200 cm3 min-1 (STP), leading to a collection of 6 L of air in each cartridge using the automatic sampler. Additional sampling was performed at 24 m with a GSA SG-10-2 personal sampler pump during the years 2012-2014. These earlier samples were collected in the same type of adsorbent tubes as for the automatic sampler, and were filled at 167 cm3 min-1 (STP) air flow for 20 min. These additional measurements took place on 19 and 28 November 2012; 1, 3 and 4 March 2013; 11 to 14 June 2013; 22, 25 and 26 September 2013 and on 17 and 21 August 2014."

1- L224 – In Table 1 it is not clear what the tolerance is.

2- We have added this information to the table legend.

1- L128 – The "Instrument for chemical analysis". Are you intent to describe the instruments or the methods used?

2- We intent to describe the methods used, therefore we have changed the name of this section to "Instruments used for chemical analysis".

1- L137 – What is "rapidly" in this context?

2- We wanted to say that the temperature increased at a high rate. We have removed this term from the text.

1- L160 – Why just 12 and 24 if all the data are available?

2- We only wanted to use 12 and 24m data of the PTR-MS as these were the only heights were GC-FID sampling took place in parallel.

1- L164 – "$2\sigma$ of the background" – $3\sigma$ it is more acceptable. Also briefly describe the blanks in each sampling systems.

2- We have modified the LOD to $3\sigma$. A more detailed explanation of the blanks has

been added as following:

3- L148: "Hourly background measurements with a catalytic converter (Supelco, Inc. with platinum pellets heated to >400°C) and weekly humidity dependent calibrations of the PTR-MS were performed" L153: "The compounds measured were monoterpenes (m/z 137) and isoprene (m/z 69). The limit of detection (LOD) of the PTR-MS for total monoterpenes was 0.1 ppb and 0.2 ppb for isoprene, determined as $3\sigma$ of the background noise."

1- L239 - Isoprene! Not mentions in material and methods. How is this measured?

2- We understand we missed this part in the materials and methods. Isoprene was measured by the PTR-MS and by GC-FID. We have added this to the text.

1- L253 – Figure not clear. E.g. isoprene 24m not visible after 12:00. Legend not descriptive, not explained what are the error bars. Some error bars below zero and yet, above LOD? Explain?

2- This figure has been modified (see attached figure 2) Now isoprene mixing ratios as measured by the GC-FID can be better seen. Furthermore, and particularly for the low mixing ratio compounds, it is possible that the measurement is 0, as it can be seen for a-terpenene, but it is not always 0. The data expressed here is an hourly average over three days and samples were collected every hour for 3 days, so having this in mind, standard deviations may be high. On the other hand, LOD for the GC-FID system is around 0.2 ppt, and this is generally way below than measured mixing ratios of the monoterpene species.

3- L737: "Figure 2: Average diel cycles for $\alpha$-pinene (a), limonene (b), myrcene (c), -cymene (d), $\beta$-pinene (e) and $\alpha$-terpinene (f) mixing ratios for 24 m (dashed line) and 12 m (thick line). In the back, average diel cycle of isoprene mixing ratios as measured by the GC-FID are shown for 24 m (light green) and 24 m (dark green). Error bars represent the standard deviation of the averages."

[Figure]

1- L345– "..in previous campaigns.. earlier samples were collected using a GSA SG10-2 personal pump sampler. Adsorbent tubes were filled at 167 cm3 min-1 (STP) air flow for 20 min". This is again an example of poor structure. This need to be in the Material and Methods section.

2- We agree with the reviewer this explanation was not well structured. We have put this information in the methods paper and have removed this from the results.

3- L113: "Furthermore, additional sampling was performed at 24 m with a GSA SG-10-2 personal sampler pump during the years 2012-2014. These earlier samples were collected in the same type of adsorbent tubes as for the automatic sampler, and were filled at 167 cm3 min-1 (STP) air flow for 20 min. These additional measurements took place on 19 and 28 November 2012; 1, 3 and 4 March 2013; 11 to 14 June 2013; 22, 25 and 26 September 2013 and on 17 and 21 August 2014."

1- L380 - How the above-canopy O3 concentrations used in the model and for the reactivity calculation represent real situation in the canopy (between 12 and 24m)?

2- We have used measured ozone mixing ratios at 12 and 24m for our calculations and not the above canopy ozone mixing ratios. We have removed this part from the results as it was already specified in the methodology and we clarified that the ozone mixing ratio levels used for reactivity calculations were at 12 and 24m. The revised text reads as:

3- L191: "The simulations with MLC-CHEM were constrained with the observed surface layer net radiation (above the canopy only), wind speed, relative humidity and O3 mixing ratios as well as the temperature measured above and inside the canopy (8 different heights including 12 and 24m) from 17 to 20 October 2015, coinciding with the measurement dates." L310: "For ozone reactivity calculations, 12 ppb was used, as this mixing ratio was observed during the measurement period. NO3 mixing ratios were taken from the MLC-CHEM model simulations that predicted mixing ratios of ∼0.4 ppt."

[Figure]

1- L439 and on – "The emission of monoterpenes has generally been thought to be from storage glands in specialized structures like resin ducts, glandular trichomes or related structures (Schürmann et al., 1993; Steinbrecher, 1989)." This generalization and discussion based further in the text are related only to conifer type or monoterpene emitters (see also in your references). Thus, this is irrelevant (and incomparable) for the tropical forest as the physiology (and chemotypes) for monoterpene emission in broadleaf tree species is different to the pool emitters.

2- We agree with the reviewer that this is an unspecific generalization and we have revised the text accordingly:

3- L260: "In contrast to plant species of cooler climates, such as spruce, which emit terpenes from pools (Ghirardo et al., 2010; Lerdau et al., 1997), Amazonian plant species have been found to show an emis-sion dependency on light and temperature (Bracho-Nunez et al., 2013; Jardine et al., 2015; Kuhn et al., 2002, 2004). This could partly explain the diurnal pattern of $\alpha$-pinene mixing ratios, which exhibit some relation to a light and temperature dependent emission flux (Kuhn et al., 2002; Rinne et al., 2002; Williams et al., 2007). However, this behaviour is not observed for all monoterpene species. Therefore, the observed diurnal cycles of some monoterpene species might be related to a stronger temperature response. While monoterpenes are stored in leaves and their release from these pools is governed by leaf temperature (Monson et al., 1995), Amazonian plant species have been found to show an emis-sion de-pendency on light and temperature (Bracho-Nunez et al., 2013; Jardine et al., 2015; Kuhn et al., 2002, 2004). This could partly explain the diurnal pattern of $\alpha$-pinene mixing ratios, which exhibit some relation to a light and temperature dependent emis-sion flux (Kuhn et al., 2002; Rinne et al., 2002; Williams et al., 2007). However, this behaviour is not observed for all monoterpene species. Therefore, the observed diur-nal cycles of some monoterpene species might be triggered by stronger temperature dependencies."

1- L502 and 522 – Any leaf level experiments to address this? Needs a brief discussion.

2- We assume that the reviewer is referring to L520-522. Unfortunately, we do not have any leaf level experiments to address this. We have performed these calculations which give an estimation of the possible role of NO in SOA growth indicating the need for further studies. We only want to point out the necessity of assessing this role when studying SOA growth from monoterpene species in the Amazon region.

Technical corrections:

1- L1 – Two words "Amazon" in a title are not needed.

2- We have removed Amazon from the title.

1- L19 – Why just in Amazon rainforest? You may just say "a rainforest".

2- We have changed it accordingly.

1- L24 – "automatic" - automated?

2- We have changed it accordingly.

1- L31 – "may not be the most atmospheric chemically relevant compounds". Although it might be grammatically correct, this is a bit odd and does not read well. This is repeated later in the text.

2- We have changed this sentence to: "Reactivity calculations showed that higher abundance does not imply higher reactivity"

1- L74-76 – "scarce" – a bit over-repetition of this word throughout the manuscript and here.

2- We have modified the wordings accordingly.

1- L230 – The is a discussion in the "Result" section.

2- We have now merged results with discussion.

1- L253 – Figure 2 legend and axes text too small.

2- We have modified this figure and increased legend and axes.

1- L283 – Figure 3 – figure caption above not needed. Description needs to be extended.

2- We have modified this figure and elaborated further in the description:

3- "Pie charts representing the day (a and c) and night (b and d)day and night averaged monoterpene species abundance in aver-age percentage with standard deviation at 24 (a and b) and 24 (c and d) m. Day period was from 0900h to 1700h and night period was from 2000h to 0500h."

1- L289 – "O1D" ?!

2- We agree with the reviewer this was not specified in the text and we have removed it.

1- L312 – Figure 4 out of margins, figure caption not needed. Use a) b): : : to refer to the individual figures with a clear description. Also, check all the figures to meet this standard.

2- We agree with the reviewer and we have modified the graph and footnote accordingly.

1- L381-384 - Repetition

2- We have deleted this part in the results sections to avoid repetition.

Ref.: Bonn, B., Bourtsoukidis, E., Sun, T. S., Bingemer, H., Rondo, L., Javed, U., Li, J., Axinte, R., Li, X., Brauers, T., Sonderfeld, H., Koppmann, R., Sogachev, A., Jacobi, S. and Spracklen, D. V.: The link between atmospheric radicals and newly formed particles at a spruce forest site in Germany, Atmos. Chem. Phys., 14, 10823–10843, doi:10.5194/acp-14-10823-2014, 2014.

Hummel, M.: Laborstudie zum Beitrag organischer Peroxyradikale (RO2) bei der Partikelneubildung während der Ethen–Ozon–Reaktion, Master's thesis, J. W. Goethe-University, Frankfurt (Main), 2010.

Sarrafzadeh, M., Wildt, J., Pullinen., I., Springer, M., Kleist, E., Tillmann, R., Schmitt, S.H., Wu, C., Mentel, T.F., Zhao, D., Hastie, D.R., and Kiendler-Scharr, A.: Impact of NOx and OH on secondary organic aerosol formation from ïĄć-pinene photooxidation. Atmos. Chem. Phys., 16, 11237–11248, 2016

Wildt, J., Mentel, T. F., Kiendler-Scharr, A., Hoffmann, T., Andres, S., Ehn, M., Kleist, E., Müsgen, P., Rohrer, F., Rudich, Y., Springer, M., Tillmann, R., and Wahner, A.: Suppression of new particle formation from monoterpene oxidation by NOx. Atmos. Chem. Phys., 14, 2789–2804, doi:10.5194/acp-14-2789-2014, 2014.ç

Wolf, J. L., Suhm, M., and Zeuch, T.: Suppressed particle formation by kinetically controlled ozone removal: revealing the role of transient-species chemistry during alkene ozonolysis, Angew. Chem., 48, 2231–2235, 2009.

Wolf, L., Richters, S., Pecher, J., and Zeuch, T.: Pressure dependent mechanistic branching in the formation pathways of secondary organic aerosol from cyclic-alkene gas-phase ozonolysis, Phys. Chem. Chem. Phys., 13, 10952–10964, 2011.

[Figure]

M. Sarrafzadeh et al.: Impact of $NO_x$ and OH on SOA formation

study
al and
wever,
he ini-
$OC]_0$,

ion in
oxida-
of seed
nstant
nizing
$g L^{-1}$)
l then
ng the
:d wa-
l con-
article
erosol

[Figure]

**Figure 1.** Measured SOA yield from $PM_{max}$ (black circles) and rates of new particle formation (blue squares) for the $\beta$-pinene photooxidation as a function of the ratio of the initial hydrocarbon to the initial $NO_x$ concentration and as a function of the initial $NO_x$ concentration. Each point corresponds to one experiment. The errors in nucleation rate and $[NO_x]$ were estimated to be $\pm 10\%$. The error in SOA yield was estimated from error propagation using the sum of the systematic error, correction procedure error and error in BVOC data. Note that the horizontal error bars are associated with the BVOC / $NO_x$ axis.

ion, a

**Fig. 1.**

[Figure]

**Fig. 2.**

---

## Author Response (AR1)

- 1 acp-2017-817
- 2 Monoterpene chemical speciation in the Amazon tropical rainforest: variation with season, height, and
- 3 time of day at the Amazon Tall Tower Observatory (ATTO)
- 4 Combined document with detailed responses to the reviewer and a copy of the revised manuscript with5 all changes tracked.
- 6
- 7
- 8 Response to reviewers 1 and 2.
- 9 1- Comments from reviewers
- 10 2- Author's response
- 11 3- Change in the text
- 12 Reviewer #1 (Remarks to the Author):
- 13 General:

14 The paper presents in-canopy, speciated BVOC measurements by GC-FID over a period of three days. These are compared PTR measurements and to longer time series of GC measurements in different sea-15 sons at the same site. The data is analyzed and discussed in terms of emission height, diurnal cycle, re-16 activity, and seasonality in comparison to the longer data series in different seasons. Observations are 17 compared to a canopy model which contained detailed chemistry for some species and rudimentary 18 chemistry for new others. The purpose of the comparison is to demonstrate speciated measurements are 19 20 a good test if the overall key features and processes are captured by the model. The manuscript presents interesting data and data analysis. The text is well structured. The manuscript could be published in 21 22 ACP after considering the following major and minor comments.

- 1- My major concern relates to the discussion of the role of deposition on wet leaf surfaces. From
   the presented material I do not see proof from the model comparison that deposition on wet
   leafs is a significant process. This may be in parts owed to the fact that description and interpre tation are not detailed enough. The authors may address the following questions and issues.
- 27 2- We understand the concern of the reviewer. We use a canopy exchange model which has been previously extensively evaluated for different ecosystems including tropical rainforest. We have 28 29 further stressed this feature of model evaluation and have included some further references for 30 more detailed information about some of the canopy model features relevant to the presented subject of BVOC exchange, e.g., estimating canopy wetness as a function of RH. We are not 31 claiming that deposition to leaf wet surfaces is a significant process. One would actually indeed 32 33 expect this sink to be of minor relevance for the terpenes included in this study given their low solubility's. However, the inferred wet vegetation uptake resistance following the widely-used 34

Wesely (1989) approach of  $\sim 300$  s m-1 results in simulation of an apparent not ignorable sink 35 also given the large area of potential wet surface. Using this inferred uptake rate we achieved a 36 relative good agreement between model simulated and observed monoterpene mixing ratios, not 37 only in magnitude (which is no surprise given the fact that the constant basal emission factor has 38 been selected to reproduce the campaign average mixing ratios) but especially regarding the di-39 urnal dynamics. The results point to a combined effect of the potential role of chemistry and 40 deposition as monoterpene sinks pointing to the necessity to explore further this deposition to 41 42 wet surface process.

43

46

**1- (I think a paragraph should be started in line 406 where the leaf wetness is discussed.)**

- 45 2- We have separated a paragraph for sink processes.
- 47 1. Did you observe the leaf surface wetness in the two nights? If yes, was it the same or was it different?
- 2- Unfortunately we did not have measurements of leaf surface wetness available for the site, and
  therefore we used relative humidity as a proxy for the calculation of leaf surface wetness. MLCCHEM uses relative humidity as a proxy for the fraction of the wet leaf surface (Lammel, 1999).
  This results in smaller estimates of canopy wetness on 17 October 2015 compared to the following days. In the revised version, we now discuss that these estimates of canopy wetness cannot
  be corroborated without measured canopy wetness.
- You suggest that the model has a strong missing sink in the night from the 17. to the 18. Is this
  the only possible explanation for the strongly overpredicted a-pinene mixing ratio? Or could be
  there more reasons for the model showing so high a-pinene in the night from the 17. to the 18.?
  Only if the source and the chemical sink of a-pinene are about the same in the two nights then
  the depositions sink must have been also similar as the mixing ratios are about the same. Were
  source and chemical sink the same in both nights?
- 2. In response to this comment we modified the discussion about the comparison of the simulated 61 and observed temporal variability in monoterpene mixing ratios and the role of canopy wetness. 62 Rather than stressing these contrasts between the night of 17-18 October and the other nights we 63 now contrast more generally the 17th and the other three days. This approach was chosen due to 64 missing meteorological observations for the 18th. There is a significant change in the modeled 65 canopy wetness conditions around midnight due to using a prescribed typical diurnal cycle in 66 some meteorological parameters instead of the actual observations for the 17th. It provides 67 though an interesting sensitivity analysis since as soon as the imposed canopy wetness increases 68 to a value of 1,  $\alpha$ -pinene drops to much lower values. This strong dependence of the simulated 69 temporal variability on changes in canopy wetness conditions is confirmed by an analysis of the 70 process tendencies in the model which show that the emissions do not significantly change com-71 paring the 17th with the other 3 days of the campaign. The contribution by chemical oxidation 72

- 73also shows some temporal variability due to differences in  $O_3$ , OH and  $NO_3$  but this sink is rela-74tively small compared to the dominant sink of  $\alpha$ -pinene, canopy deposition. There the changes in75canopy wetness contrasting the 17th with the other 3 days explains to a large extent the simulated76temporal variability in mixing ratios of  $\alpha$ -pinene and the other terpenes and which generally77agrees quite well with the observed temporal variability.
- How can deposition on wet leaves affect so strongly the mixing ratio of non-water soluble compounds like a-pinene?
- 2- What we present here, relies on application of inferred uptake efficiencies for these monoter-80 penes not only commonly being used in any model study following the Wesley (1989) approach 81 to consider dry deposition but also recently being applied in a detailed study on Boreal forest 82 canopy exchange of BVOCs (Zhou et al., 2017). The inferred monoterpene wet-surface uptake 83 resistances on the order of 300 s m-1, based on Henry law's constants on the order of  $\sim 1-3^{e-2}$  M 84 atm-1, suggest a potentially still quite efficient uptake by leaf wetness. Note that this seems to be 85 similar to the observed removal of ozone by wet canopies where, despite its low solubility, there 86 is actual experimental evidence that canopy wetness enhances O3 deposition potentially due to 87 aqueous-phase chemical interactions. Clearly, this feature on leaf wetness in monoterpene re-88 moval has to be corroborated by further experimental evidence. 89
  - 1- I understand there was no RH data for the 18. How could the model derive a reliable surface wetness then?
  - 2- See also our response to the previous points about missing canopy wetness observations and the impact of the imposed meteorological data for the 18th of October.
- Why do the different MT respond so differently to the ill predicted surface wetness on 17.: a pinene and b-pinene too high, limonene ok, myrcene in between, and terpinolene too low. Inso far I think the statement in line 527 is not justified by the presented material and analysis.
- 97 2- In the revised version we have excluded the respective statement.
- 98 Major comments:

92

- line 514-518: I do not understand what the authors are discussing here. I cannot understand how
   a "potential" aerosol growth could have a dimension of molecules per cm3 and an order of 10-5
   -10-6. Moreover, I did not find such notations in Bonn et al. 2014. This part must be clarified or
   deleted.
- 103 2- Our aim was to discuss the effects of mono- and sesquiterpenes as classified in the Amazon on particle formation rates i.e. not aerosol growth rates. Admiringly, the reviewer is correct as the 104 wrong wording was used. The units provided are wrong as the unit needs to be "cm-3 s-1". These 105 have been corrected in the revision. A particle growth would focus on size change per time i.e. 106 nm h-1 for a certain initial size but not change concentrations per time as provided. In general, 107 we used the word "potential" as no direct measurements of particle growth or of particle size 108 distribution were available and calculations are based on the transfer of results from elsewhere 109 to Brazil. With respect to the aerosol formation and growth rate, it has been shown (for example 110

- by Wolf et al. (2009: 2011) and by Hummel (2010) and included in Bonn et al. (2014)) that the 111 initial growth of (smaller) particles is determined by organic peroxy radicals as one of two pro-112 cesses. Those processes are (a) the reactive interaction of large organic peroxy radicals (RO2) 113 from e.g. mono- and sesquiterpene reactions with OH or NO3 and (b) in case sufficient organic 114 mass has been acquired already uptake by partitioning of oxidized VOCs with a reduced volatil-115 ity. Both processes can be simulated and large organic RO2 concentrations approximated assum-116 ing steady-state conditions for radicals which is a common assumption. For clarification the text 117 has been changed to: 118
- 3- L388 (please refer to supplement for correct location of line number): "Following the equation 119 established by Bonn et al. (2014) (Equation number 5 in text), we were able to estimate the po-120 tential aerosol particle number formation rate initiated by monoterpene species only  $(1 \times 10^{-5} \text{ to})$ 121  $5x10^{-5}$  cm-3 s-1 at 24 m) assuming steady state conditions for radicals. Those were found to be 122 approximately two orders of magnitude smaller than the calculated potential new aerosol parti-123 cle formation rate caused by oxidation products of sesquiterpenes. Our calculations assume mix-124 ing ratios of sesquiterpenes of 0.2 ppb revealing potential formation rates of  $1 \times 10^{-3}$  and  $4.5 \times 10^{-3}$ 125 cm-3 s-1 at 24 m) based on previous measurements in the Amazon (Jardine et al., 2011) which are 126 remarkably smaller than observed at mid-latitude conditions (Bonn et al., 2014)." 127
- 128
- line 519f: This a strong statement. Do you have proof for that or a reference? Moreover, potential aerosol growth and formation rate are not the same. Indeed NO suppresses nucleation (Wildt et al. ACP, 2014) but not so much the yield (Sarrafzadeh et al. ACP, 2016)
- We kindly refer the reviewer to Sarrafzadeh et al. (2006) and the respective Figure 1 that illustrates how SOA yield depends on the BVOC/NOx ratio (see attached figure 1).
- 134For Amazonian conditions (in contrast to smog chamber measurement conditions) the typical
- BVOC mixing ratios are smaller than 3 ppb and most of the time around 1 ppb. If we apply Fig. 1 of Sarrafzadeh et al. (2016), the situation is located on the very left of the plot. Increasing NOx 1 at constant BVOC mixing ratio will decrease the BVOC/NOx ratio and lead to a decline in SOA 1 yield. The unaffected region of the same figure is not applicable for the Amazon region because 1 of present concentrations. This is in line with the findings of Wildt et al. (2014) for smaller parti-1 cles indicating similar processes responsible for the growth. For clarification we provide with a 1 more detailed explanation as below:
- L396: "Furthermore, the level of present NO (nitric oxide) also affects the potential aerosol growth (Wildt et al., 2014) and yield (Sarrafzadeh et al., 2016) at low BVOC/NOx ratios. As the theory assumes contributions of larger organic peroxy radicals (RO2), which are destroyed by reactions e.g. with NO, increasing NOx at constant BVOC mixing ratio will decrease the BVOC/NOx ratio and lead to a decline in SOA yield. Our calculations showed this effect, with a change of NO from 0.2 ppby to 1 ppby leading to a decrease in the formation rate at a diameter
  - 4

- of 3 nm. This interdependence calls for a consistent consideration of the BVOC and NOx exchange in aerosol formation and growth studies."
- 150 Minor comments:
- 151 1- Starting with line 219 the authors give errors/uncertainty ranges/standard deviations in form of 152 "+/-yx" at many instances (including Table 1) without explaining the specific meaning.
- **153** 2- We have added averages +/- standard deviation where corresponded in the text.
- 154 1- line 156: "142 Td", Td is not explained
- 155 2- This is now explained in the text.
- line 186f: The formulation "selected" suggests to me that the initial EA and the fractional EA for
   individual compounds were used as adaptable parameters. Is this case? If so, then the overall
   good agreement between modelled and observed concentrations in section 3.5 (line 398f) and
   Figure 6 is not surprising. If not, explain the in more detail the rationale for these selections.
- 2- The emission factors from the selected monoterpenes were indeed adaptable parameters. We 160 used selected constant basal emissions for terpenes for the plant ecosystem class, in this case 161 tropical rainforest. We used initially basal emission factor for monoterpenes for tropical rainfor-162 est of 0.4 ug C g-1 hr-1 (Guenther et al., 1995). Then we partitioned this emission flux over the 163 different monoterpene species to see how well the model would reproduce the observed mixing 164 ratios. Following, we adjusted the value for each species until the observed 4-day average mix-165 ing ratios were reasonably well reproduced. Given the daily changes in the processes potentially 166 involved in explaining the observed temporal variability, the presented analysis gives you infor-167 mation about the role of the different processes, as these have different dominating effects at dif-168 ferent times of the day and depend, at the same time, on many other drivers. 169
- 170In the text we have chosen to show the basal emissions in mg C  $g^{-1} h^{-1}$ , rather than emission fac-171tor constant, and we have modified the text as follows in order to better explain this issue.
- 3- L179: "The monoterpene basal leaf-scale emission factors have been selected as such that the 172 model simulates monoterpene mixing ratios of comparable magnitude compared to the cam-173 paign average observed mixing ratios. In the evaluation of simulated and observed mixing ratios 174 we mainly focus on the comparison between the simulated and observed temporal variability be-175 ing determined by the differences in canopy processes for contrasting nocturnal and daytime 176 conditions. For the model simulation, the basal emission factors were 0.18  $\mu$ g C g-1 h-1 for  $\alpha$ -pi-177 nene, 0.04  $\mu$ g C g-1 h-1 for  $\beta$ -pinene, 0.11  $\mu$ g C g-1 h-1 for  $\alpha$ -terpinene, 0.9  $\mu$ g C g-1 h-1 for limo-178 nene and 0.18  $\mu$ g C g-1 h-1 for myrcene. Note the selected relative high basal emission flux for 179 limonene is required to reach simulated mixing ratios comparable to the observed ones." 180
- 1- line 386f: Similar is true for the formulation about the "assumed" lead area index and leaf area density. What is the rationale for these choices? Did you estimate it from observations on the site?
- LAI was previously measured at the site, and we have included this reference in the methodol ogy.

- line 256, first §: These percentages are averages over day and night hours and three measurement days? They should have standard deviations.
- 188 2- We agree with the reviewer and have added the standard deviations of the percentages both in189 the graph and text.
- line 286, first §: I suggest to define what you mean by reactivity. e.g. OH reactivity = kOH\*[MT].
- **192** 2- We have added this clarification in the text.
- 1- line 303, Table 2: The header of the third column is misleading. It should read "normalized reac tivity for 1 ppb [s-1]", or so. Otherwise confusion with the use of reactivity later in the text.
- **195** 2- We have modified this in Table 2 accordingly.
- 196 1- line 489: Is Hallquist 2009 (a review) a good reference for this statement?
- 197 2- We agree this is not the most appropriate reference and we have excluded it.
- line 542f: That is not new and with the given formulation the conclusion in its generality does
   not make sense. The authors did not show that there is no suited representation of MT, which
   considers also their reactivity. Moreover, the degree of tolerable simplification depends also on
   the purpose of the model calculation.
- 202 2- We have reformulated the sentence, and we believe the point of the sentence was not well stated
  203 before.
- 1- L502 "Furthermore, reactivity calculations demonstrated that higher abundance of MT does not 204 automatically imply higher reactivity as the most abundant compounds may not be the most at-205 206 mospheric chemically relevant compound or the relative contribution of different monoterpenes may change. Our calculations support the view to that the role of canopy exchange may be erro-207 neously estimated when not taking into account speciation based reactivity in models. Moreover, 208 simulations with a canopy exchange modelling system to assess the role of canopy interactions 209 compared relatively well with the observed temporal variability in speciated monoterpenes, but 210 also indicate the necessity of more experiments to enhance our understanding of in-canopy sinks 211 of these monoterpenes." 212
- 213 Reviewer #2 (Remarks to the Author):
- 214 General comments:
- 1- The manuscript "Monoterpene chemical speciation in the Amazon tropical rainforest: variation with season, height, and time of day at the Amazon Tall Tower Observatory (ATTO)" is a suitable and in the scope of "Atmosphere Chemistry and Physics", shading light on the importance of monoterpene characterisation. Although, the manuscript is interesting it requires a major revision as it lacks the clear structure. The reader is often forced to go forward and beck. Many discussions are found in Results (e.g. L230-232, L241-242, L267-270, L276-278, L280-281 etc.).
- 221 Many statements are started with not suitable paragraph context (e.g. L306: "The most abundant

- species are  $\alpha$ -pinene, limonene and myrcene".). Some statements and discussion are out of con-
- text or even incorrect as mentioned in the Specific comments below. It is also too long for the scientific content. I suggest the following: Restructure the text in order to join Results and Discussion into one section with adequate subsections, redo the figures as suggested below and de-
- 226 lete the repetitions in the text. Please also find specific and technical comments.
- 2- We thank the reviewer for the fruitful feedback. As suggested, we have now merged the results
   and discussion sections, removed repetitive sentences and redid the figures.
- 229 Specific comments:
- L60-63. Not entirely true. PTR-MS could have a time resolution <1s (not 30 s). It is also recently coupled with FastGC to characterise monoterpenes in < 2min. 1h GCFID is a bit too much for "current" method. Please update the references for all this.</li>
- 233 2- We agree with the reviewer and therefore we have removed that part from the text.
- L103 and 125 Not indicated sampling frequency, 30 min mentioned in L125, is not in agree ment with the resolution presented in the Figure 1b and c. A detailed but structured description
   of the methods used is needed.
- 2- The sampling frequency was every hour for 30 minutes. We agree with the reviewers this is not
   clear in the text and we have modified this paragraph for better explanation.
- 239 3- L111 "The samples were collected from 17 to 20 October 2015 at an hourly frequency. Samples
  240 were collected for 30 min every hour at a flow of 200 cm3 min-1 (STP), leading to a collection of
  241 6 L of air in each cartridge."
- L106 Air sampling section needs structure improvement. You first open with your sampler
   giving the reference, then you describe your sampling dates/times and then you come back to the
   sampler, and again back to the sampling procedure.
- 2- We thank the reviewer for this comment, as it is true that section needed improvements. There-fore, we have restructured this section following:
- 3- L97: "Collection of ambient air samples on adsorbent tubes, for subsequent analysis by Gas 247 248 Chromatography – Flame Ionization Detector (GC-FID), was made with two automated cartridge samplers described earlier (Kesselmeier et al., 2002; Kuhn et al., 2002, 2005) positioned 249 at 12 and 24 m on the INSTANT tower. The samplers consist of two main units, a cartridge 250 magazine that holds the adsorbent-filled tubes and the control unit timing the process and re-251 cording the data. This latter unit also houses the pumps (Type N86KT, KNF Neuberger, Frei-252 burg, Germany), pressure gauges, mass flow controllers and power supply. The cartridge maga-253 zine is equipped with solenoid valves controlling the inlet and outlet of up to 20 individual sam-254 255 pling adsorbent tubes. The system is a constant-flow device, with one cartridge position per loop used as a bypass for purging the system. Due to the compact weatherproof housings and the low 256 power consumption, we were able to position one sampler at 24 m and the other one at 12 m, 257 attached to the INSTANT tower booms with commercially available 50 mm aluminium clamps. 258 The adsorbent tubes used for VOC sampling were filled with 130 mg of Carbograph 1 (90 m2 g- 259

- 1) followed by 130 mg of Carbograph 5 (560 m2 g-1) sorbents. The size of the Carbograph parti-260 cles was in the range of 20–40 mesh. Carbographs 1 and 5 were provided by L.A.R.A s.r.l. 261 (Rome, Italy) (Kesselmeier et al., 2002). The samples were collected from 17 to 20 October 262 2015. Samples were taken for 30 min every hour at a flow of 200 cm3 min-1 (STP), leading to a 263 collection of 6 L of air in each cartridge using the automatic sampler. Additional sampling was 264 performed at 24 m with a GSA SG-10-2 personal sampler pump during the years 2012-2014. 265 These earlier samples were collected in the same type of adsorbent tubes as for the automatic 266 sampler, and were filled at 167 cm3 min-1 (STP) air flow for 20 min. These additional measure-267 ments took place on 19 and 28 November 2012; 1, 3 and 4 March 2013; 11 to 14 June 2013; 22, 268 25 and 26 September 2013 and on 17 and 21 August 2014." 269
- 1-L224 In Table 1 it is not clear what the tolerance is.
- 271 2- We have added this information to the table legend.
- L128 The "Instrument for chemical analysis". Are you intent to describe the instruments or the methods used?
- 2- We intent to describe the methods used, therefore we have changed the name of this section to
  "Instruments used for chemical analysis".
- 276 1- L137 What is "rapidly" in this context?
- 277 3. We wanted to say that the temperature increased at a high rate. We have removed this term from the text.
- 1- L160 Why just 12 and 24 if all the data are available?
- 2- We only wanted to use 12 and 24m data of the PTR-MS as these were the only heights were
   GC-FID sampling took place in parallel.
- L164 "2σ of the background" 3σ it is more acceptable. Also briefly describe the blanks in
   each sampling systems.
- 2- We have modified the LOD to 3σ. A more detailed explanation of the blanks has been added as following:
- 286 3- L148: "Hourly background measurements with a catalytic converter (Supelco, Inc. with platinum pellets heated to >400°C) and weekly humidity dependent calibrations of the PTR-MS were performed"
- L153: "The compounds measured were monoterpenes (m/z 137) and isoprene (m/z 69). The
  limit of detection (LOD) of the PTR-MS for total monoterpenes was 0.1 ppb and 0.2 ppb for isoprene, determined as 3σ of the background noise."
- 1- L239 Isoprene! Not mentions in material and methods. How is this measured?
- 2- We understand we missed this part in the materials and methods. Isoprene was measured by the
   PTR-MS and by GC-FID. We have added this to the text.
- L253 Figure not clear. E.g. isoprene 24m not visible after 12:00. Legend not descriptive, not
   explained what are the error bars. Some error bars below zero and yet, above LOD? Explain?
- 297 2- This figure has been modified (see attached figure 2)

- Now isoprene mixing ratios as measured by the GC-FID can be better seen. Furthermore, and particularly for the low mixing ratio compounds, it is possible that the measurement is 0, as it can be seen for a-terpenene, but it is not always 0. The data expressed here is an hourly average over three days and samples were collected every hour for 3 days, so having this in mind, standard deviations may be high. On the other hand, LOD for the GC-FID system is around 0.2 ppt, and this is generally way below than measured mixing ratios of the monoterpene species.
- 304 3- L737: "Figure 2: Average diel cycles for α-pinene (a), limonene (b), myrcene (c), ρ-cymene (d),
   305 β-pinene (e) and α-terpinene (f) mixing ratios for 24 m (dashed line) and 12 m (thick line). In the
   306 back, average diel cycle of isoprene mixing ratios as measured by the GC-FID are shown for 24
   307 m (light green) and 24 m (dark green). Error bars represent the standard deviation of the averages."
- 309
- L345- "..in previous campaigns.. earlier samples were collected using a GSA SG10-2 personal
   pump sampler. Adsorbent tubes were filled at 167 cm3 min-1 (STP) air flow for 20 min". This is
   again an example of poor structure. This need to be in the Material and Methods section.
- 313 2- We agree with the reviewer this explanation was not well structured. We have put this information in the methods paper and have removed this from the results.
- 315 3- L113: "Furthermore, additional sampling was performed at 24 m with a GSA SG-10-2 personal sampler pump during the years 2012-2014. These earlier samples were collected in the same type of adsorbent tubes as for the automatic sampler, and were filled at 167 cm3 min-1 (STP) air flow for 20 min. These additional measurements took place on 19 and 28 November 2012; 1, 3 and 4 March 2013; 11 to 14 June 2013; 22, 25 and 26 September 2013 and on 17 and 21 August 2014."
- L380 How the above-canopy O3 concentrations used in the model and for the reactivity calculation represent real situation in the canopy (between 12 and 24m)?
- We have used measured ozone mixing ratios at 12 and 24m for our calculations and not the
   above canopy ozone mixing ratios. We have removed this part from the results as it was already
   specified in the methodology and we clarified that the ozone mixing ratio levels used for reactiv ity calculations were at 12 and 24m.
- 327 The revised text reads as:
- 3- L191: "The simulations with MLC-CHEM were constrained with the observed surface layer net radiation (above the canopy only), wind speed, relative humidity and O3 mixing ratios as well as the temperature measured above and inside the canopy (8 different heights including 12 and 24m) from 17 to 20 October 2015, coinciding with the measurement dates."
- 332 L310: "For ozone reactivity calculations, 12 ppb was used, as this mixing ratio was observed
- during the measurement period. NO3 mixing ratios were taken from the MLC-CHEM model
- simulations that predicted mixing ratios of ~0.4 ppt."

- L439 and on "The emission of monoterpenes has generally been thought to be from storage glands in specialized structures like resin ducts, glandular trichomes or related structures (Schürmann et al., 1993; Steinbrecher, 1989)." This generalization and discussion based further in the text are related only to conifer type or monoterpene emitters (see also in your references). Thus, this is irrelevant (and incomparable) for the tropical forest as the physiology (and chemotypes) for monoterpene emission in broadleaf tree species is different to the pool emitters.
- 341 2- We agree with the reviewer that this is an unspecific generalization and we have revised the text342 accordingly:
- 3- L260: "In contrast to plant species of cooler climates, such as spruce, which emit terpenes from 343 pools (Ghirardo et al., 2010; Lerdau et al., 1997), Amazonian plant species have been found to 344 show an emis-sion dependency on light and temperature (Bracho-Nunez et al., 2013; Jardine et 345 al., 2015; Kuhn et al., 2002, 2004). This could partly explain the diurnal pattern of  $\alpha$ -pinene 346 mixing ratios, which exhibit some relation to a light and temperature dependent emission flux 347 (Kuhn et al., 2002; Rinne et al., 2002; Williams et al., 2007). However, this behaviour is not ob-348 served for all monoterpene species. Therefore, the observed diurnal cycles of some monoterpene 349 350 species might be related to a stronger temperature response. While monoterpenes are stored in leaves and their release from these pools is governed by leaf temperature (Monson et al., 1995). 351 Amazonian plant species have been found to show an emission de-pendency on light and tem-352 perature (Bracho-Nunez et al., 2013; Jardine et al., 2015; Kuhn et al., 2002, 2004). This could 353 partly explain the diurnal pattern of  $\alpha$ -pinene mixing ratios, which exhibit some relation to a 354 355 light and temperature dependent emission flux (Kuhn et al., 2002; Rinne et al., 2002; Williams et al., 2007). However, this behaviour is not observed for all monoterpene species. Therefore, 356 357 the observed diurnal cycles of some monoterpene species might be triggered by stronger temperature dependencies." 358
- 1- L502 and 522 Any leaf level experiments to address this? Needs a brief discussion.
- We assume that the reviewer is referring to L520-522. Unfortunately, we do not have any leaf
   level experiments to address this. We have performed these calculations which give an estima tion of the possible role of NO in SOA growth indicating the need for further studies. We only
   want to point out the necessity of assessing this role when studying SOA growth from monoter pene species in the Amazon region.
- 365 Technical corrections:
- 366 1- L1 Two words "Amazon" in a title are not needed.
- **367** 2- We have removed Amazon from the title.
- 368 1- L19 Why just in Amazon rainforest? You may just say "a rainforest".
- **369 2-** We have changed it accordingly.
- 1- L24 "automatic" automated?
- **371** 2- We have changed it accordingly.

| 372        | 1-            | L31 – "may not be the most atmospheric chemically relevant compounds". Although it might be                                                                                                                                                       |  |  |  |  |  |  |
|------------|---------------|---------------------------------------------------------------------------------------------------------------------------------------------------------------------------------------------------------------------------------------------------|--|--|--|--|--|--|
| 3/3        | 2             | We have changed this sentence to: "Peactivity calculations showed that higher abundance does                                                                                                                                                      |  |  |  |  |  |  |
| 374        | 2-            | not imply higher reactivity?                                                                                                                                                                                                                      |  |  |  |  |  |  |
| 375        | 1             | 174.76 "approx" a bit over repetition of this word throughout the manuscript and here                                                                                                                                                             |  |  |  |  |  |  |
| 376        | 1-            | L/4-/0 – scarce – a bit over-repetition of this word throughout the manuscript and here.                                                                                                                                                          |  |  |  |  |  |  |
| 3//        | Z-
1       | L 220 The is a discussion in the "Desult" section                                                                                                                                                                                                 |  |  |  |  |  |  |
| 378        | 1-            | L250 – The is a discussion in the Result section.                                                                                                                                                                                                 |  |  |  |  |  |  |
| 379        | Z-
1       | L 252 Element 2 les endenderes tert ter energil                                                                                                                                                                                                   |  |  |  |  |  |  |
| 380        | 1-            | L255 – Figure 2 legend and axes text too small.                                                                                                                                                                                                   |  |  |  |  |  |  |
| 381        | Z-
1       | We have modified this figure and increased legend and axes.                                                                                                                                                                                       |  |  |  |  |  |  |
| 382        | 1-            | L283 – Figure 3 – figure caption above not needed. Description needs to be extended.                                                                                                                                                              |  |  |  |  |  |  |
| 383        | 2-            | we have modified this figure and elaborated further in the description:                                                                                                                                                                           |  |  |  |  |  |  |
| 384        | 3-            | "Pie charts representing the day (a and c) and night (b and d)day and night averaged monoter-                                                                                                                                                     |  |  |  |  |  |  |
| 385        |               | pene species abundance in aver-age percentage with standard deviation at 24 (a and b) and 24 (c                                                                                                                                                   |  |  |  |  |  |  |
| 386        |               | and d) m. Day period was from 0900h to 1700h and night period was from 2000h to 0500h."                                                                                                                                                           |  |  |  |  |  |  |
| 387        |               |                                                                                                                                                                                                                                                   |  |  |  |  |  |  |
| 388        | 1-            | L289 – "O1D" ?!                                                                                                                                                                                                                                   |  |  |  |  |  |  |
| 389        | 2-            | We agree with the reviewer this was not specified in the text and we have removed it.                                                                                                                                                             |  |  |  |  |  |  |
| 390        | 1-            | L312 – Figure 4 out of margins, figure caption not needed. Use a) b): : : to refer to the individual                                                                                                                                              |  |  |  |  |  |  |
| 391        |               | figures with a clear description. Also, check all the figures to meet this standard.                                                                                                                                                              |  |  |  |  |  |  |
| 392        | 2-            | We agree with the reviewer and we have modified the graph and footnote accordingly.                                                                                                                                                               |  |  |  |  |  |  |
| 393        | 1-            | L381-384 - Repetition                                                                                                                                                                                                                             |  |  |  |  |  |  |
| 394        | 2-            | We have deleted this part in the results sections to avoid repetition.                                                                                                                                                                            |  |  |  |  |  |  |
|            | _             |                                                                                                                                                                                                                                                   |  |  |  |  |  |  |
| 395        | Ref.:         |                                                                                                                                                                                                                                                   |  |  |  |  |  |  |
| 396        | Bonn, B       | ., Bourtsoukidis, E., Sun, T. S., Bingemer, H., Rondo, L., Javed, U., Li, J., Axinte, R., Li, X., Brauers, T., Sonderfeld, H.,                                                                                                                    |  |  |  |  |  |  |
| 397        | k             | Coppmann, R., Sogachev, A., Jacobi, S. and Spracklen, D. V.: The link between atmospheric radicals and newly formed                                                                                                                               |  |  |  |  |  |  |
| 398        | Ŗ             | particles at a spruce forest site in Germany, Atmos. Chem. Phys., 14, 10823–10843, doi:10.5194/acp-14-10823-2014,                                                                                                                                 |  |  |  |  |  |  |
| 399
400 | Humme         | .014.
  M : Laborstudie zum Beitrag organischer Peroxyradikale (RO2) bei der Partikelneubildung während der Ethen-                                                                                                                             |  |  |  |  |  |  |
| 401        | (             | Dzon–Reaktion, Master's thesis, J. W. Goethe-University, Frankfurt (Main), 2010.                                                                                                                                                                  |  |  |  |  |  |  |
| 402        | Sarrafza      | deh, M., Wildt, J., Pullinen., I., Springer, M., Kleist, E., Tillmann, R., Schmitt, S.H., Wu, C., Mentel, T.F., Zhao, D., Hastie,                                                                                                                 |  |  |  |  |  |  |
| 403        | [             | D.R., and Kiendler-Scharr, A.: Impact of NO x and OH on secondary organic aerosol formation from $\beta$ -pinene                                                                                                                       |  |  |  |  |  |  |
| 404        | ۲
۱۸/۱۱۰۰۰ | photooxidation. Atmos. Chem. Phys., 16, 11237–11248, 2016
Montol, T. E. Kiendler Scharr, A. Hoffmann, T. Andres, S. Ehn, M. Kleist, E. Müssen, D. Behrer, E. Budich, V.                                                                        |  |  |  |  |  |  |
| 406        | vvnut, J      | WIIGT, J., IVIENTEI, I. F., KIENDIER-SCHART, A., HOTTMANN, I., ANDRES, S., ENN, M., KIEIST, E., Musgen, P., Kohrer, F., Rudich, Y.,
Springer M. Tillmann, R. and Wahner A. Suppression of new particle formation from monoterpene oxidation by |  |  |  |  |  |  |
| 407        | 1             | IOx. Atmos. Chem. Phys., 14, 2789–2804, doi:10.5194/acp-14-2789-2014, 2014.                                                                                                                                                                       |  |  |  |  |  |  |
| 408        | Wolf, J.      | L., Suhm, M., and Zeuch, T.: Suppressed particle formation by kinetically controlled ozone removal: revealing the role                                                                                                                            |  |  |  |  |  |  |
| 409        | 0             | f transient-species chemistry during alkene ozonolysis, Angew. Chem., 48, 2231–2235, 2009.                                                                                                                                                        |  |  |  |  |  |  |
| 410
411 | Wolt, L.      | , Richters, S., Pecher, J., and Zeuch, T.: Pressure dependent mechanistic branching in the formation pathways of                                                                                                                                  |  |  |  |  |  |  |
| 411        | S             | econuary organic aerosof from cyclic-arkene gas-phase ozofiolysis, Phys. Chefn. Chefn. Phys., 13, 10952–10964, 2011.                                                                                                                              |  |  |  |  |  |  |
|            |               | 11                                                                                                                                                                                                                                                |  |  |  |  |  |  |

- 413
- Monoterpene chemical speciation in the Amazona tropical rainforest: variation with season, height,
   and time of day at the Amazon Tall Tower Observatory (ATTO)
- Ana María Yáañez-Serrano1,\*, Anke Christine Nölscher1,\*\*, Efstratios Bourtsoukidis1, Eliane
  Gomes Alves2, Laurens Ganzeveld3, Boris Bonn4, Stefan Wolff1, Marta Sa2, Marcia Yamasoe5,
  Jonathan Williams1, Meinrat O. Andreae1,6 and Jürgen- Kesselmeier1

[revised manuscript text omitted]
 $\alpha$ -pinene, camphene, sabinene, $\beta$ -pinene, myrcene, $\alpha$ -phellandrene, 3-carene, $\alpha$ -terpinene, $\rho$ - |
| 565 | cymene, limonene and $\gamma$ -terpinene. Isoprene was also quantified. The detection limit for the GC-FID                                |
| 566 | was 2 ppt (Bracho-Nunez et al., 2011).                                                                                                    |
| 567 |                                                                                                                                           |
|     |                                                                                                                                           |
| 568 | 2.3.2. Proton Transfer Reaction - Mass Spectrometer (PTR-MS)                                                                              |
| 569 | Online total monoterpene mixing ratios were determined by a quadrupole Proton Transfer Reac-                                              |
| 570 | tion - Mass Spectrometer, PTR-MS (Ionicon Analytic, Austria). The PTR-MS was operated under                                               |
| 571 | standard conditions (2.2 mbar drift pressure, 600 V drift voltage, with an E/N of 142 Townsend (Td)).                                     |
| 572 | In addition to weekly humidity dependent calibrations, hourly background measurements were per-                                           |
| 573 | formed with a catalytic converter (Supelco, Inc. with platinum pellets heated to >400°C). A gravimetri-                                   |
| 574 | cally prepared multicomponent standard for calibration was obtained from Apel & Riemer Environmen-                                        |
| 575 | tal, USA. The measurements were carried out at two different heights (0.05, 0.5, 4, 12 and, 24, 53 and                                    |
| 576 | 79 m) with the PTR-MS switching sequentially between each height at 2 min intervals-and only data                                         |
| 577 | from 12 and 24 m is shown. The inlet lines were made of PTFE (9.5 mm OD), insulated and heated to                                         |
| 578 | 50 °C, and had PTFE particle inlet filters at the intake end. The compounds of interest for this study                                    |
| 579 | were the isoprene ( $m/z$ 69) and the sum of monoterpenes ( $m/z$ 137) and isoprene (in $m/z$ 69). The limit                              |
| •   |                                                                                                                                           |

of detection (LOD) of the PTR-MS for total monoterpenes was 0.1 ppband 0.2 ppb for isoprene, determined as 3σ of the background noise. More information about the gradient system and PTR-MS operation at ATTO can be found elsewhere (Nölscher et al., 2016; Yáñez-Serrano et al., 2015).

583 584

**2.4. Multi-Layer Canopy Chemistry Exchange model (MLC-CHEM)**

585 To analyse the magnitude and temporal variability inof the observed monoterpene concentrations inside and above the forest canopy, we applied the Multi-Laver Canopy Chemistry Exchange 586 Model (MLC-CHEM), driven by the observed micro-meteorology and ozone surface layer mixing ra-587 tios. MLC-CHEM was originally developed and implemented in a single-column model (SCM). It is 588 originally set up as well as also in a global chemistry and climate-modelling system to assess the role of 589 590 canopy processes in local- to global-scale atmosphere-biosphere exchange of nitrogen oxides (Ganzeveld et al., 2008, 2002; Kuhn et al., 2010). The model's generalized representation of chemistry, 591 dry deposition, emissions and turbulent mixing allows to studying the role of canopy interactions in de-592 termining atmosphere-biosphere exchange fluxes and in-canopy and surface layer mixing ratios of, e.g., 593 ozone  $(O_3)$ , nitrogen oxides  $(NO_x)$  and biogenic volatile organic compounds (BVOCs). The BVOC 594 emissions are calculated according to MEGAN (Guenther et al., 2006), considering the vertical distribu-595 596 tion of biomass and direct as well as diffuse radiation to calculate leaf-scale BVOC emissions. The current implementation of canopy chemistry in MLC-CHEM considers, in addition to standard photo--597 598 chemistry involving  $O_3$ ,  $NO_x$ , methane (CH4) and carbon monoxide (CO), the role of non-methane hydrocarbons including isoprene, and a selection of hydrocarbon oxidation products such as formalde-599 hyde, higher aldehydes and acetone. Oxidation of the monoterpenes by OH,  $O_3$  and  $NO_3$  is taken into 600 601 account, but the role of the monoterpene oxidation products in photo-chemistry is not considered in the current implementation of the chemistry scheme in MLC-CHEM. For this study, we have extended 602 MLC-CHEM to consider, besides the already included compounds  $\alpha$ -pinene and  $\beta$ -pinene, the observed 603 monoterpene species,  $\alpha$ -terpinene, limonene and myrcene. The monoterpene basal leaf-scale monoter-604 605 pene emission factors have been selected such that the model simulates monoterpene mixing ratios of comparable magnitude compared to the campaign-average observed mixing ratios. In the evaluation of 606

- 607 simulated and observed mixing ratios we mainly focus on comparison of the simulated and observed 608 temporal variability being determined by the differences in canopy processes for contrasting nocturnal and daytime conditions. For the model simulation, the basal emission factors were 0.18  $\mu$ g C g-1 h-1 for 609  $\alpha$ -pinene, 0.04 ug C g-1 h-1 for  $\beta$ -pinene, 0.11 ug C g-1 h-1 for  $\alpha$ -terpinene, 0.9 ug C g-1 h-1 for limonene 610 and 0.18 ug 
[revised manuscript text omitted]

| 699      | pinene represented more than 85% of the total monoterpeneMT mixing ratio (Figure 3). During the day                            |
| 700      | (0900h to 1700h) $\alpha$ -pinene had an average abundance (average-±standard deviation) of 46±25% and                         |
| 701      | 36±4% of the total monoterpene mixing ratios at 24 and 12 m, respectively, and was the dominant mon-                           |
| 702      | oterpene in this study overall. However, during the night (2000h to 0500h), its relative abundance                             |
| 703      | dropped to $25\pm13\%$ and $25\pm9\%$ at 24 and 12 m, respectively. In contrast, limonene made up $23\pm15\%$                  |
| 704      | and 20±10% of the monoterpenes at 24 and 12 m, respectively, by day, and increased during night time                           |
| 705      | to 33±15% and 26±16% at 24 and 12 m. Thus, there wasis a tendency of towards some differences in                               |
| 706      | monoterpene species abundances between day and night. These wereare mainly due to the nocturnal de-                            |
| 1
707 | creases in $\alpha$ -pinene and the nocturnal relative increase in limonene. It is plausible that the observed de-             |
| 708      | crease in $\alpha$ -pinene mixing ratios could be due to $\frac{1}{\alpha}$ decreased vegetation emission, as reduced chemical |
| 709      | destruction due to very low OH concentrations at night, would lead to an increase in the nocturnal $\alpha$ -                  |
| -
710 | pinene mixing ratios.                                                                                                          |

Even though there were clear differences between the absolute and relative abundances of some monoterpene species during day and night, there were no clear changes in the in the vertical gradients (e.g. for  $\alpha$ -pinene night time averages were  $0.15\pm0.05$  ppb for 12 m and  $0.11\pm0.06$  ppb at 24 m). For the day, the apparent difference in the abundance of  $\alpha$ -pinene was due to a single outlier data point covering 30 minutes at noon on 19 October 2015 at 24 m, when the  $\alpha$ -pinene mixing ratio doubled. This increase could not be explained, although it could be related to a strong change in wind speed an hour before the measurement, when the wind was blowing from the North. In general, our observations indicate that the abundance of monoterpene species does not vary much over the heights selected (12 and 24 m) within the canopy. This is consistent with the results by Kesselmeier et al. (2000), where the monoterpene composition at the rain forest floor was comparable to the above-canopy composition at their site.

722

723

**3.3. Reactivity**

724 The variability of the oxidants  $(OH, O_3 \text{ and } NO_3)$  present in the Amazon air is important when 725 considering the impact that monoterpenes can have oin the oxidative regime in the Amazon region and 726 Brazil in general. Hydroxyl radicals are produced mainly during the day via ozone photolysis. and O4D 727 reaction with water. Low levels of OH can be also generated by the reaction of ozone with doubly 728 bonded species (e.g. monoterpenes and sesquiterpenes) even at night. In this assessment, we considered 729 the monoterpene contributions to OH reactivity by day only. In contrast, NO3 is photolytically de-730 stroved during the day, but can become significant at night, so we assessed the impact of monoterpenes on NO3 reactivity at night. Even though in the Amazon rainforest ozone levels are low (~10-20 ppb) 731 732 compared to other areas of the world (e.g., Williams et al., 2016), it is nevertheless present, and some monoterpenes are extremely reactive towards ozone. Table 2 gives an overview of the lifetime and reac-733 734 tivity (which is defined as reaction rate constant (oxidant i.e. OH)\*[monoterpene species]) to 1 ppb of all the investigated monoterpene species for these three oxidants. For calculating the lifetime of the dif-735 736 ferent monoterpenes as presented in Table 2, typical oxidant concentrations for the Amazon rainforest conditions were used. For OH a mean value of  $7x10^5$  molecules cm-3 was used as representative of the 737 738 site (Spivakovsky et al., 2000). For ozone reactivity calculations, 12 ppb was used, as this mixing ratio was observed during the measurement period. NO3 mixing ratios were taken from the MLC-CHEM 739 740 model simulations that predicted mixing ratios of  $\sim 0.4$  ppt.

741

Despite, While the most abundant species were α-pinene, limonene and myrcene were the most
 abundant species, - However, with respect to their reactivities towards the different oxidants, their relative contribution to total monoterpene reactivity dramatically changed was not proportional to their

745abundances. The most abundant monoterpene,  $\alpha$ -pinene was not the dominantting sink for the oxidants.746For instanceIn particular,  $\alpha$ -terpinene dominated ozone reactivity associated with monoterpene abun-747dance both during the day and night, as well as the nocturnal nitrate reactivity, despite the low mixing748ratios measured for this compound (Table 2).

749

The monoterpene ozone reactivity wasis comparable between day  $(1.37 \times 10^{-6} \text{ s}^{-1})$  and night  $(1.12 \times 10^{-6} \text{ s}^{-1})$ 750 1). α-Tterpinene dominated the monoterpene-ozone chemistry, followed by myrcene and limonene. De-751 752 spite the relatively high abundance of  $\alpha$ -pinene (46±25%; average mixing ratio and standard deviation during the day was  $0.34\pm0.04$  ppb at 12 m), its contribution to ozone reactivity with respect to other 753 754 monoterpene species was only  $11\pm7$  % and  $3\pm1$  % at 24m, during the day and night, respectively, at 24 755  $m_{\tau}$  and  $2\pm 1$  % for both day and night at 12 m (Figure 4). As previously noted, the differences in ozone 756 reactivity between heights were negligible for the night and slightly higher at 24 m during the day. As ozone mixing ratios are quite similar for both heights during day and night (11.4 ppb at 12 m and 10.4 757 ppb at 24 m during night, and 16.1 ppb at 12 m and 15.6 at 24 m during the day), the higher abundance 758 759 of  $\alpha$ -pinene during the day, and the lower  $\alpha$ -terpinene mixing ratios at 24 m during the day mainly ex-760 plain these changes in monoterpene-ozone reactivity. It is important to note that these results are de-761 rived from a relative abundance analysis, and unmeasured monoterpene species could change the pro-762 portions, although given the close similitude between PTR-MS and GC-FID measurements shown in Figure 1 this is unlikely. On the other hand, very reactive species which could dominate reactivity, may 763 764 be present in very low concentrations, and which our measurements capabilities would not allow for its 765 monitoringbeing detected.

| 766

The monoterpene reactivity towards the NO3 radical during the night in this study was also dominated by  $\alpha$ -terpinene (40±36% and 42±27%, respectively for 24 and 12 m, respectively), although contributions of limonene (30±13% and 25±14%, respectively for 24 and 12 m, respectively),  $\alpha$ -pinene (11±6 and 11±4%, respectively for 24 and 12 m, respectively), and myrcene (13±11 and 16±12%, re- <del>spectively</del> 
[revised manuscript text omitted]
 varia-894 bility as well as the model simulations considering monoterpene emissions that only depend on temper-895 896 ature.

897

The generally quite good agreement between the simulated and observed monoterpene mixing 898 ratios, except of an overestimation of simulated  $\alpha$ -pinene mixing ratios for the 17th of October. ex-899 presses the overall result of temporally varying emissions, in-canopy chemistry, turbulent mixing and 900 deposition. The latter also involves a potentially important role of deposition to wet leaf surfaces (the 901 inferred wet surface uptake resistances for the monoterpenes are ~300 s m-1, similar to values reported 902 by Zhou et al., (2017), MLC-CHEM uses relative humidity as a proxy for the fraction of the leaf surface 903 being wet, (Lammel, 1999; Sun et al., 2016). This results in substantially smaller estimates of canopy 904 905 wetness on 17 October 2015-compared to the following days, which partly explains the simulated high

| 906 | <math>\alpha</math>-pinene mixing ratios . The simulated $\alpha$ -pinene mixing ratios for 18-20 th -of October, with inferred wet |
|-----|-------------------------------------------------------------------------------------------------------------------------------------------------------|
| 907 | surface fractions up to 1 during the night and ~0.5 during daytime, are in much better agreement with                                                 |
| 908 | the observations. Regarding the comparison of the simulated observed mixing ratios for some of the                                                    |
| 909 | other monoterpenes, the simulated ß-pinene, limonene, and myrcene mixing ratios, especially at 12.5 m                                                 |
| 910 | seem to capture the observed temporal variability quite well. Note that this result for limonene reflects                                             |
| 911 | the use of a high leaf basal emission factor (0.9 µg C g -1 hr -1 ) required to simulate mixing ratios reach-                   |
| 912 | ing up to 0.4 ppb. These MLC-CHEM simulations were also used to infer how much of the actual emis-                                                    |
| 913 | sion flux escapes the canopy, expressed by the calculated atmosphere-biosphere limonene flux divided                                                  |
| 914 | by the canopy emission flux of limonene. This ratio reaches a maximum value of 0.5 around noontime,                                                   |
| 915 | implying that these model simulations indicate that at the middle of the day, about 50% of the emitted                                                |
| 916 | limonene is removed inside the canopy by in-canopy oxidation -and deposition. During night time, this                                                 |
| 917 | ratio reaches a minimum < 0.1 indicating simulation of very efficient in-canopy removal.                                                              |

919 These modelling results should be interpreted with carecaution, also given that some of the sim-920 ulated processes cannot be evaluated due to missing observations of canopy wetness as well as the uptake efficiency of monoterpenes by wet surfaces. It should be considered that the simulated removal of 921 922 monoterpenes by wet canopy surfaces could also compensate for a misrepresentation of other canopy 923 processes, e.g., reduced emissions from wet canopy surfaces or an underestimation of the oxidation effi-924 ciency. Further analysis of the model simulated process tendencies (Ganzeveld et al., 2008) indicates 925 only small changes in the simulated source of the monoterpenes over the 4-day period. Regarding the 926 sink of, for example,  $\alpha$ -pinene, chemical destruction of  $\alpha$ -pinene oxidation by O3, OH and NO3 appears to be a relative small term, with the overall sink being dominated by deposition to wet surfaces showing 927 quite large temporal variability. Consequently, the presented quite reasonable agreement between simu-928 lated and observed temporal variability in monoterpenes mixing ratios indicates that deposition to wet 929 930 surfaces may play an important role in monoterpene atmosphere-biosphere exchange. -This should be 931 further corroborated, calling for experiments to determine the actual efficiency (and mechanisms) of up-932 take of monoterpenes by wet canopy surfaces.

**933 4. Conclusions**

| 934 | This study presents an analysis of the measured monoterpene chemodiversity at the Amazon                  |
|-----|-----------------------------------------------------------------------------------------------------------|
| 935 | tropical forest measurement site, ATTO. The results showed a distinctly different chemical speciation     |
| 936 | between day and night, whereas there were little vertical differences in speciation within the canopy (12 |
| 937 | and 24 m). Furthermore, inter- and intra-annual results demonstrate similar chemodiversity during the     |
| 938 | dry seasons analysed, but this a change of chemodiversity changed with season, similar to the seasonal    |
| 939 | measurements performed by Kesselmeier et al. (2002). Furthermore, reactivity calculations demon-          |
| 940 | strated that higher abundance of a monoterpene species MT does not automatically imply higher reac-       |

[revised manuscript text omitted]
        | $0.33\pm0.04$   | 0.15±0.05         | 0.38±0.21         | 0.11±0.06       |  |
| Limonene        | 0.18±0.09       | $0.18 \pm 0.10$   | 0.19±0.12         | $0.14 \pm 0.07$ |  |
| Myrcene         | 0.16±0.14       | $0.12 \pm 0.09$   | $0.09 \pm 0.04$   | $0.07 \pm 0.06$ |  |
| P-Cymene        | $0.07 \pm 0.03$ | $0.04 \pm 0.01$   | $0.08 \pm 0.04$   | $0.04 \pm 0.02$ |  |
| β-Pinene        | 0.08±0.03       | $0.06 \pm 0.03$   | $0.05 \pm 0.03$   | $0.04 \pm 0.02$ |  |
| Camphene        | 0.03±0.03       | $0.02 \pm 0.01$   | $0.03 \pm 0.02$   | $0.01 \pm 0.01$ |  |
| α-Terpinene     | $0.03 \pm 0.02$ | $0.03 \pm 0.02$   | $0.01 \pm 0.02$   | $0.02 \pm 0.02$ |  |
| γ-Terpinene     | $0.02 \pm 0.01$ | $0.01 \pm 0.01$   | $0.01 \pm 0.01$   | $0.01 \pm 0.01$ |  |
| 3-Carene        | 0.001±0.003     | $0.003 \pm 0.008$ | $0.003 \pm 0.011$ | 0 or BLD        |  |
| α-Phellandrene  | 0 or BLD        | 0 or BLD          | 0 or BLD          | 0 or BLD        |  |
| Sabinene        | 0 or BLD        | 0 or BLD          | 0 or BLD          | 0 or BLD        |  |
| MT Sum – GC-FID | 0.91±0.10       | 0.62±0.19         | 0.82±0.34         | 0.45±0.13       |  |
| MT Sum – PTR-MS | 0.96±0.27       | $0.54 \pm 0.17$   | $0.77 \pm 0.22$   | $0.56 \pm 0.16$ |  |
|                 |                 |                   |                   |                 |  |

**1162** Table 2: Lifetime of the different monoterpene species related to OH, O3 and NO3 for the OH daytime conditions at 24 m and at 12 m. In addition, the normalized reactivity to 1 ppb of the different monoterpene species is calculated.

| Monoterpenes investi- | Formula        | Lifetime (minutes) |                       |        | Normalized r Reactivity to 1
ppb s -1 |                       |                 |
|-----------------------|----------------|--------------------|-----------------------|--------|------------------------------------------------------------|-----------------------|-----------------|
| gated                 |                | OH                 | O 3 | $NO_3$ | OH                                                         | O 3 | NO 3 |
| α-Pinene              | $C_{10}H_{16}$ | 449                | 615                   | 250    | 1.42                                                       | 2.3E-06               | 0.17            |
| Camphene              | $C_{10}H_{16}$ | 447                | 57422                 | 2461   | 1.43                                                       | 2.4E-08               | 0.02            |
| Sabinene              | $C_{10}H_{16}$ | 400                | 623                   | 155    | 1.60                                                       | 2.2E-06               | 0.27            |
| β-Pinene              | $C_{10}H_{16}$ | 320                | 3445                  | 618    | 2.00                                                       | 4.0E-07               | 0.07            |
| Myrcene               | $C_{10}H_{16}$ | 71                 | 110                   | 141    | 8.98                                                       | 1.3E-05               | 0.30            |
| α-Phellandrene        | $C_{10}H_{16}$ | 132                | 17                    | 21     | 4.84                                                       | 8.1E-05               | 1.96            |

| ∆3-Carene   | $C_{10}H_{16}$                | 271  | 1397   | 170    | 2.37 | 9.9E-07 | 0.24    |
|-------------|-------------------------------|------|--------|--------|------|---------|---------|
| α-Terpinene | $C_{10}H_{16}$                | 103  | 2      | 11     | 6.24 | 5.6E-04 | 3.76    |
| ρ-Cymene    | $C_{10}H_{14}$                | 1577 | >90000 | >90000 | 0.41 | 1.3E-09 | 2.7E-05 |
| Limonene    | $C_{10}H_{16}$                | 145  | 246    | 127    | 4.41 | 5.6E-06 | 0.33    |
| γ-Terpinene | $C_{10}H_{16}$                | 140  | 369    | 53     | 4.57 | 3.8E-06 | 0.78    |
| Isoprene    | C 5 H 8 | 238  | 4069   | 238    | 2.69 | 3.4E-07 | 0.02    |